# Barley Leaf Insoluble Dietary Fiber Alleviated Dextran Sulfate Sodium-Induced Mice Colitis by Modulating Gut Microbiota

**DOI:** 10.3390/nu13030846

**Published:** 2021-03-05

**Authors:** Meiling Tian, Daotong Li, Chen Ma, Yu Feng, Xiaosong Hu, Fang Chen

**Affiliations:** College of Food Science and Nutritional Engineering, National Engineering Research Center for Fruit and Vegetable Processing, Key Laboratory of Fruit and Vegetables Processing Ministry of Agriculture, Engineering Research Centre for Fruits and Vegetables Processing, Ministry of Education, China Agricultural University, Beijing 100083, China; tml0214@163.com (M.T.); lidaotong@bjmu.edu.cn (D.L.); machen21@sina.com (C.M.); fengyu9459@163.com (Y.F.); huxiaos@263.net (X.H.)

**Keywords:** barley leaf, insoluble dietary fiber, gut microbiota, secondary bile acid

## Abstract

Supplementation of dietary fiber has been proved to be an effective strategy to prevent and relieve inflammatory bowel disease (IBD) through gut microbiota modulation. However, more attention has been paid to the efficacy of soluble dietary fiber than that of insoluble dietary fiber (IDF). In the present study, we investigated whether IDF from barley leaf (BLIDF) can inhibit gut inflammation via modulating the intestinal microbiota in DSS-induced colitis mice. The mice were fed 1.52% BLIDF-supplemented diet for 28 days. Results demonstrated that feeding BLIDF markedly mitigated DSS-induced acute colitis symptoms and down-regulated IL-6, TNF-α, and IL-1β levels in the colon and serum of colitis mice. BLIDF supplementation effectively reduced the abundance of *Akkermansia* and increased the abundance of *Parasutterella*, *Erysipelatoclostridium*, and *Alistipes*. Importantly, the anti-colitis effects of BLIDF were abolished when the intestinal microbiota was depleted by antibiotics. Furthermore, the targeted microbiota-derived metabolites analysis suggested that BLIDF feeding can reverse the DSS-induced decline of short-chain fatty acids and secondary bile acids in mice feces. Finally, BLIDF supplementation elevated the expression of occludin and mucin2, and decreased the expression of claudin-1 in colons of DSS-treated mice. Overall, our observations suggest that BLIDF exerts anti-inflammatory effects via modulating the intestinal microbiota composition and increasing the production of microbiota-derived metabolites.

## 1. Introduction

Inflammatory bowel disease (IBD), including Crohn’s disease and ulcerative colitis, is a chronic recurrent intestinal immune-mediated disease characterized by dysregulated immune responses and gut microbiota [1,2]. Numerous studies demonstrate that intestinal microbiota dysbiosis is highly associated with IBD development [3]. The intestinal microbiota in IBD hosts is characterized by a decline in microbial diversity and altered bacterial composition, including a reduced abundance of *Clostridia*, *Lachnospiraceae*, and *Alistipes*, and an elevated abundance of Enterobacteriaceae species [4,5]. Consistently, colitis developed in genetically susceptible mice with a conventional microbiota rather than germ-free colony [6]. Moreover, oral transfer of the fecal microbiota from IBD patients into microbiota-depleted *IL-10*^−/−^ mice promoted colitis development [7]. These results indicate that the intestinal microbiota is a crucial factor in colonic inflammation development. Therefore, targeting microorganisms is a novel strategy to intervene the pathogenesis and development of IBD.

Accumulating evidence suggests that dietary fiber, especially soluble dietary fiber, is an effective regulator in shaping host gut microbiota against the development of IBD [8,9,10]. Soluble dietary fiber, such as oligofructose-enriched inulin and fructans, exerted beneficial outcomes in patients with IBD by elevating gut *Bifidobacterium* abundance [9]. Resistant starch was reported to increase the numbers of *Faecalibacterium prausnitzii*, *Ruminococcus bromii*, *Parabacteroides distasonis*, and *Clostridium leptum*, helping to mitigate gut inflammation [10]. Moreover, dietary fiber altered gut microbiota and contributed to the increase of short-chain fatty acids (SCFAs) such as acetate, propionate, and butyrate [11]. Propionate and butyrate exert anti-inflammatory effects on intestinal epithelial cells, dendritic cells, and macrophages, via inhibiting histone deacetylases activity [12]. Butyrate also exhibits anti-colitis effects via activating G protein-coupled receptor 43 in mice [13]. IDF is the major component of plant cell wall, which is mainly composed of cellulose, hemicellulose, and lignin [14]. There is consensus that the major physiological benefit of insoluble dietary fiber (IDF) is its fecal bulking effect, as it is less fermented by the gut bacteria in the colon compared with soluble dietary fiber [15,16]. Interestingly, it is recently reported that cellulose supplementation attenuates dextran sodium sulfate (DSS)-induced mice colitis, and a correlation between the cellulose-induced increase in *Akkermansia muciniphila* and colitis alleviation was found [17,18]. However, it is not clear whether the IDF extracted from plants can alleviate colitis via gut microbes.

Barley (*Hordeum vulgare* L.) leaves are rich in IDF (53.6 g/100 g dry weight) and has an extended history of usage in traditional Chinese medicine because of its benefits on gastrointestinal functions, anti-inflammatory effects, and hypolipidemic effects [19]. Our previous study demonstrated that barley leaf greatly enhanced the growth of fiber-degrading bacteria (Lachnospiraceae and *Prevotella*) and the SCFA (propionate and butyrate) production in mice [19]. This finding inspired us to investigate whether barley leaf insoluble dietary fiber (BLIDF) can prevent inflammation by regulating gut microbiota in colitis mice. In the current work, we investigated the influence of supplementation of BLIDF on colitis symptoms, gut microbiota composition, microbiota-derived metabolites, and intestinal barrier function, in DSS-treated mice. Our work suggested that dietary supplementation of BLIDF prominently ameliorated DSS-induced gut microbiota dysbiosis, which may result in the increased production of microbiota-derived metabolites, restoration of impaired gut barrier function, and remission of colitis. This work is of great significance in evaluating the efficacy of IDF on intestinal microbiota modulation in DSS-induced colitis mice and providing fundamental evidences for the further application of IDF in IBD interventions.

## 2. Materials and Methods

### 2.1. Preparation of BLIDF

Barley leaf (the same batch used in the previous study) was obtained from Hebei Biotechnology Co., Ltd. (Jiaxing, Zhejiang, China) and prepared as described previously [19]. IDF of barley leaf was extracted by the enzymatic method. Briefly, barley leaf powder (5 g) was suspended in distilled water (200 mL) and continuously stirred using a JJ-1 digital display motor stirrer (Jintan Instruments Co., Ltd., Jiangsu, China) at 100 °C for 35 min. The mixture was adjusted to pH 7.0 after cooling and incubated with 160 U/mL of neutral protease (200,000 U/g, Solarbio Bioscience &Technology Co., Ltd. Beijing, China) at 50 °C for 40 min with continuous stirring. Afterward, the mixture was filtered through Whatman No. 1 filter paper using a vacuum filtration unit (Shanghai Zhenjie Experimental Equipment Co., Ltd., Shanghai, China). The residue was sequentially washed with 70 °C distilled water, 95% ethanol, acetone, and distilled water, three times each. The washed residue was lyophilized using a vacuum freeze-dryer (LGJ-18, Beijing Songyuan Huaxing Biotechnology Co., Ltd., Beijing, China), smashed, sieved (60-mesh sieve), and the final powder was stored in a desiccator until use.

The protein, fat, and moisture contents of BLIDF were determined based on the Association of Official Analytical Chemists (AOAC) 978.04, 960.39, and 934.01 methods, respectively. The results were shown in Appendix A. The cellulose, hemicellulose, and lignin contents of BLIDF were determined based on the NREL method [20] and the results were shown in Appendix A. The monosaccharide composition of BLIDF was determined by high-performance liquid chromatography with precolumn derivatization using PMP (1-phenyl-3-methyl-5-pyrazolone) [21]. The chromatograms were shown in Appendix A.

### 2.2. Animal Diets

The control diet was a standard chow diet. The BLIDF diet was prepared by adding BLIDF powder to the control diet at a ratio of 1.52%. The ratio was calculated by transforming a dose of mouse daily intake of IDF (in the form of whole barley leaf) to an equivalent dose of pure BLIDF [19]. The composition of the two diets, provided by Huafukang Biotechnology Co., Ltd. (Beijing, China), is listed in Appendix A.

### 2.3. Animal Experiments

Five-week-old female C57BL/6J mice were purchased from Beijing Vital River Laboratory Animal Technology Co., Ltd. (Beijing, China) and housed at a standard specific-pathogen-free animal facility with a 12-h light and dark cycle. the grouping, feeding, and treating procedures are clarified as follows.

For experiment I, after adaptation for 7 days, 24 mice were randomly divided into three groups: CT group, CT+DSS group, BLIDF+DSS group, 4 mice per cage. The CT and CT+DSS groups were fed a control diet, and the BLIDF+DSS group was fed with BLIDF diet throughout the experiment. After feeding *ad libitum* for 21 days, 2.5% (*w*/*v*) DSS (36,000–50,000 kDa; MP Biomedicals, Solon, OH, USA) in drinking water was administered for 7 days (DSS water was changed every 2 days) in the CT+DSS and BLIDF+DSS groups.

For experiment II, after adaptation for 7 days, 16 mice were treated with an antibiotics mixture (ab) containing ampicillin, metronidazole, and neomycin (1 g/L; Sigma–Aldrich, St. Louis, MO, USA) and vancomycin (500 mg/L; Sigma–Aldrich) in drinking water for 14 days and divided into CT+DSSab and BLIDF+DSSab groups, 4 mice per cage. The ab-containing water was changed every 3 days. After 14 days of the ab treatment, DSS was added to the ab-containing water (2.5% *w*/*v*) (water was changed every 2 days) and administrated for 7 days. Group CT+DSSab were fed a control diet, and group BLIDF+DSSab were fed BLIDF diet throughout the experiment.

In all the animal studies, water and food intake were recorded at each time point and the information was shown in Appendix A. Animals were weighed in the morning of each day in the last week of the experiment. At the end of the experiment, the blood sample was collected from the posterior orbital venous plexus, incubated at room temperature, and the serum was separated (1000× *g*, 10 min). The mice were killed by cervical spine dislocation. The cecal contents were collected for microbiota, SCFAs, and bile acids analysis. Colon segments were divided into four parts, one part was fixed with 10% formalin and used for histological analysis. The other three parts were used for inflammatory cytokines analysis, targeted mRNA expression, and western blot analysis, respectively.

### 2.4. Disease Activity Index (DAI) and Histological Assessment

The mice of all groups were observed from the beginning of DSS treatment to the end of the experimental investigation, quantified using DAI evaluations based on the clinical symptoms of weight loss, diarrhea, and bloody stool [22]. These indices were scored on a scale ranging from 0 (no disease symptoms) to 4 (severe disease symptoms).

Formalin-fixed colonic segments were embedded in paraffin, carefully sectioned, and stained with hematoxylin and eosin (H&E) (Servicebio Technology Co., Ltd., Hubei, China). The stained sections were then blindly scored for histological assessment on a scale of 1–12 according to the previous description [23].

### 2.5. Inflammatory Cytokines Analysis

Enzyme-linked immunosorbent assay (ELISA) kits were used to measure the con-centrations of TNF-α, IL-6, IL-1β, IL-10, and IL-4 in serum and colon tissue lysates fol-lowing manufacturer’s instruction. Briefly, 50 mg colon tissue was mixed with 500 μL phosphate buffer saline (PBS, pH 7.4, Solarbio Bioscience & Technology Co., Ltd., Beijing, China) and was homogenized using Omni bead ruptor 12 (Omni International, Kennesaw, GA, USA). Colon tissue lysates were separated (12,000× *g*, 4 °C, 15 min) and their protein concentrations were analyzed using the BCA protein assay kit (Solarbio Bioscience & Technology Co., Ltd., Beijing, China). ELISA kits for TNF-α (ab208348), IL-6 (ab100712), and IL-1β (ab197742) were obtained from Abcam (Cambridge, MA, USA). Those for IL-10 (VAL605) and IL-4 (VAL603) were bought from R&D Systems (Minneapolis, MN, USA).

### 2.6. Quantitative Real-Time PCR (qRT-PCR)

Total RNA was extracted from colon tissues of mice with RNeasy Plus Mini Kit (Qiagen, Hilden, Germany) according to the manufacturer’s instructions. An equal amount of RNA (1.0 μg) was transcribed into complementary DNA (cDNA) using Transcriptor Reverse Transcriptase (Takara Bio, Inc., Shiga, Japan). The qRT-PCR analysis was performed on a LightCycler^®^ 480 Real-Time PCR System (Roche Diagnostics Co., Basel, Switzerland) using protocols described previously [24]. The expression levels of *occludin*, *claudin-1*, *claudin-3*, and *mucin2* were normalized to *GAPDH* expression and quantified using the 2^−ΔΔ*Ct*^ method. The primer sequences used in this study were listed in Appendix A.

### 2.7. 16S rRNA Amplicon Sequencing

16S rRNA gene sequencing analysis was performed by Majorbio Bio-Pharm Technology Co., Ltd. (Shanghai, China) as previously described [16]. According to the manufacturer’s instructions, the genomic DNA of cecal contents was extracted using the QIAamp-DNA stool mini kit (Qiagen, Hilden, Germany). The V3-V4 hypervariable regions of the 16S rRNA gene were amplified using the genomic DNA of cecal content as the template and the primers (338F: 5′-GTGCCAGCMGCCGCGG-3′, 806R: 5′-CCGTCAATTCMTTTRAGTTT-3′) that target conserved sequences found in bacteria [16]. The PCR amplification products were sequenced on an Illumina MiSeq platform (Majorbio Bio-Pharm Technology Co., Ltd., Shanghai, China). After quality filtration and merger using Fastp software, the 16S rRNA gene sequencing data were classified into operational taxonomic units (OTUs) with 97% similarity. OTU representative sequences were assigned taxonomy using the SILVA database (Release 128 http://www.arb-silva.de, accessed on 26 March 2019). Analyses for rarefaction curves and alpha diversity (Ace, Chao1, Shannon, and Simpson indexes) were calculated with the MOTHUR program. The principal coordinate analysis (PCoA) plot was generated in QIIME 1 using the Chord distance matrix and statistically analyzed using PERMANOVA. Differences in the relative abundance of bacteria between groups were analyzed by the Wilcoxon rank-sum test at False Discovery Rate (FDR) < 0.05. The difference between groups was considered to be significant with FDR-corrected *p*-value < 0.05. All these data statistics were performed on the free online Majorbio I-Sanger Cloud Platform (www.i-sanger.com, accessed on 26 March 2019). All raw sequence data has been deposited at the NCBI Short Read Archive database under the BioProject accession number PRJNA700444 (data will be released at 8 March 2022).

### 2.8. SCFAs Analysis

The freeze-dried and grounded cecal contents samples were mixed thoroughly with 800 μL of deionized H_2_O and 200 μL of 50% H_2_SO_4_. The mixture was added to 1 mL of diethyl ether and shaken by vortex for 2 min, then left in ice for 10 min. The supernatant was collected after centrifugation (12,000× *g*, 10 min) and dehydrated by anhydrous CaCl_2_.

The resulting supernatant was analyzed on a GC2010 Plus system (Shimadzu Co., Tokyo, Japan) equipped with a Stabilwax-DA fused silica capillary column (30 m × 0.32 mm × 0.50 μm). A flame ionization detector was used with the injector temperature at 260 °C, followed by temperature programming of holding 80 °C for 1.5 min, increasing to 240 °C at a rate of 10 °C/min, holding at 240 °C for 20 min. Nitrogen was used throughout the measurement process as a carrier gas. The concentration of acetate, propionate, butyrate, isobutyrate, valerate, and isovalerate were calculated based on the peak area of standard samples (Sigma–Aldrich, St. Louis, MO, USA).

### 2.9. Bile Acids Quantification

Bile acids were quantified by the Shanghai Biotree Biotech Co., Ltd. (Shanghai, China). Briefly, cecal content bile acids were isolated in extracting solvent containing distilled water (20%), acetonitrile (40%), methanol (40%), formic acid (0.1%), and 250 nmol/L of the internal standards (taurochenodeoxycholic acid (TCDA)-[d4], glycocholic acid-[^2^H_4_], and deoxycholic acid (DCA)-[d6]). All samples were mixed by vortex (30 s), sonicated (5 min), and placed in a –40 °C ice bath (1 h). The supernatants were separated (12,000× *g*, 4 °C, 15 min) and analyzed by UHPLC-QE Orbitrap/MS. The separation and analysis conditions of the UHPLC-QE Orbitrap/MS were described in previous literature [25].

### 2.10. Western Blot

Colon segments were homogenized and sonicated in RIPA buffer with PMSF, proteinase, and phosphatase inhibitors. The protein extracts were centrifuged (12,000× *g*, 15 min) to remove tissue debris, separated by electrophoresis on 12% sodium dodecyl sulfate-polyacrylamide gel (SDS-PAGE), and transferred to PVDF membranes. The membranes were blocked with 5% skim milk and incubated with primary antibodies, such as claudin-1 (51-9000, Invitrogen, Carlsbad, CA, USA), occludin (Invitrogen), and GAPDH (#2118, Cell Signaling Technology, Danvers, MA, USA) at 4 °C overnight, and then incubated with the corresponding secondary antibodies (Huaxingbio Co., Ltd., Beijing, China) for 1.5 h at room temperature. The membranes were washed three times for 10 min each, incubated with SuperSignal chemiluminescent substrate (Huaxingbio Co., Ltd., Beijing, China), and imaged by Amersham Imager 600 (General Electric Company, Pittsfield, MA, USA). Blots were semi-quantified using ImageJ software (National Institutes of Health, Bethesda, MD, USA).

### 2.11. Immunohistochemistry

Following deparaffinization with xylene, colonic sections were rehydrated with sodium citrate antigen repair solution (Solarbio Bioscience & Technology Co., Ltd., Beijing, China). The colonic sections were then combined with 3% H_2_O_2_ to block the endogenous peroxidase activity and washed three times in PBS before incubating with primary antibody mucin2 (ab97386, Abcam, Inc. Cambridge, MA, USA) at 4 °C overnight. The colonic sections were washed with PBS three times and then incubated with corresponding secondary antibodies (Huaxingbio Co., Ltd., Beijing, China) at room temperature for 2 h. Finally, diaminobenzidine (DAB) chromogenic liquid was added to the slices for color development. The expression levels of mucin2 were quantitated by integral optical density (IOD) using image pro plus 6.0 (Media Cybernetics, Rockville, MD, USA).

### 2.12. Statistical Analysis

Data are expressed as mean ± SEM. Unless otherwise stated, the difference between groups was analyzed by IBM SPSS software (20.0 version, IBM Corporation, Armonk, NY, USA) using one-way analysis of variance (ANOVA) with least significant difference (LSD) (equal variances assumed) or Tamhane’s T2 test (equal variances not assumed). The difference between groups was considered to be significant at *p* < 0.05. Correlation analyses were calculated using MetaboAnalyst 5.0 (https://www.metaboanalyst.ca/, accessed on 11 January 2021) and plotted using TBtools (https://github.com/CJ-Chen/TBtools/releases, accessed on 2 November 2020). The other figures were plotted using GraphPad Prism software (7.0 version, Graphpad Software, La Jolla, CA, USA).

## 3. Results

### 3.1. BLIDF Supplementation Mitigated DSS-Induced Colitis in Mice

The protective effects of BLIDF against colitis were evaluated in DSS-induced acute colitis in mice (Figure 1A). As expected, a significant body weight loss, elevated DAI score, and colon length shortening were observed in the CT+DSS group (Figure 1B–E). Moreover, inflammatory lesions, including significant and complete goblet cell loss, crypt distortion, and inflammatory cell infiltration, were found in the CT+DSS group (Figure 1F,G). Notably, the weight loss, DAI scores, colon length shortening, and inflammatory lesions in DSS-treated mice were significantly decreased by BLIDF supplementation (Figure 1B–G). These results demonstrate that BLIDF supplementation strikingly alleviated DSS-induced colitis.

### 3.2. BLIDF Decreased Pro-Inflammatory Cytokine Levels in DSS-Treated Mice

Inflammatory cytokines act as signaling molecules to mediate the inflammatory response in various immune cells [26]. Raised levels of inflammatory cytokines can reflect the severity of gut inflammation to some extent [26]. To evaluate whether the anti-colitis effects of BLIDF were associated with an improvement in the inflammatory responses, several most-detected inflammatory cytokines were quantified in serum and colonic tissue by ELISA. DSS induced a significant increase of pro-inflammatory cytokines (IL-6, TNF-α, and IL-1β) in both serum and colonic tissue of mice, while BLIDF supplementation significantly reduced their levels in DSS-treated mice (Figure 2A,B). Moreover, BLIDF supplementation dramatically attenuated the DSS-induced elevated expression of anti-inflammatory cytokines IL-4 and IL-10 in colonic tissue (Figure 2B). The down-regulation of pro-inflammatory and anti-inflammatory cytokine levels in colitis alleviated mice is consistent with a previous study [27].

### 3.3. The Colitis Alleviation Effects of BLIDF Supplementation Depend on Gut Microbiota

Intestinal microbiota plays a critical role in IBD therapeutics and pathogenesis [28,29]. In the current work, 16S rDNA sequencing was used to determine whether BLIDF can alter the gut bacterial composition in DSS-treated mice. From the 24 cecal contents samples, a total of 1,081,553 high-quality sequences and an average of 45,064.71 ± 48,46.93 sequences per sample were obtained. After quality filtering and removal of chimeric sequences, singletons, and doubletons, 593 operational taxonomic units (OTUs) were identified. Rarefaction analysis indicated that adequate sequencing depth, and most bacterial diversity, were captured in the samples (Appendix A). DSS and BLIDF administration has no significant effect on Ace and Chao index (Figure 3A,B), suggesting the unchanged gut microbiota richness between groups. However, the increased Shannon index and decreased Simpson index of the DSS-treated mice indicated a significant increase in their gut microbiota α-diversity, which was not reversed by BLIDF feeding (Figure 3C,D). Nevertheless, the principal coordinate analysis (PCoA) based on Chord distances displayed significant difference (PERMANOVA, 999 permutations, R^2^ = 0.12601, *p* = 0.049) in the intestinal bacterial composition between CT+DSS and BLIDF+DSS groups (Figure 4A), suggesting that BLIDF altered the intestinal microbiota composition in colitis mice.

We further analyzed the influences of BLIDF on the composition of intestinal bacteria at the phylum and genus levels. At the phylum level, the top seven abundant bacterial phyla, including Firmicutes, Bacteroidetes, Proteobacteria, Actinobacteria, Verrucomicrobia, Deferribacteres, and Saccharibacteria, were identified (Figure 4B, Appendix A). DSS administration led to a decline in the abundance of Firmicutes and Actinobacteria and growth of Bacteroidetes, Proteobacteria, and Verrucomicrobia compared to the CT group (Figure 4B). Notably, BLIDF feeding remarkably decreased the proportion of Verrucomicrobia in colitis mice without significantly influencing the other bacterial phyla (Figure 4C, Appendix A). At the genus level, DSS administration led to a remarkable increase in the abundances of *Akkermansia*, *Parasutterella*, and *Erysipelatoclostridium* and a reduction in *Alistipes* (Figure 4D,E) as compared to control. With BLIDF supplementation, the relative abundances of *Parasutterella*, *Erysipelatoclostridium*, and *Alistipes* were sharply elevated by 3.3-, 4.0-, and 5.2-fold, and the relative abundance of *Akkermansia* declined dramatically by 5.3-fold as compared to DSS mice, respectively (Figure 4D,E). However, BLIDF supplementation did not significantly prevent the increase of *Escherichia-Shigella*, *Bacteroides*, *Parabacteroides*, *Mucispirilum*, *Ruminiclostridium*, *norank_f__Rhodospirillaceae*, *norank_f__Ruminococcaceae*, *Ruminiclostridium_9*, *Anaerotruncus*, *Enterococcus*, and *[Ruminococcus]_torques_group,* and the decrease of *Faecalibaculum*, *Lactobacillus*, *Bifidobacterium*, and *Lachnoclostridium* (Appendix A). We also examined the relationship between the above four significantly influenced genera and the DAI score, colon length, pathological score, and the levels of five inflammatory cytokines (IL-6, TNF-α, IL-1β, IL-4, and IL-10) in all groups of mice. Spearman’s correlation analyses (Figure 4F) showed that *Akkermansia* abundance was highly positively associated with DAI score, pathological score, IL-6, TNF-α, IL-1β, IL-4 and IL-10, and highly negatively associated with colon length. *Alistipes* abundance was highly negatively associated with DAI score, pathological score, IL-6, TNF-α, IL-1β, and IL-10, while highly positively associated with colon length. *Parasutterella* and *Erysipelatoclostridium* abundance exhibited significant positive correlations with DAI score, pathological score, TNF-α, IL-1β and IL-4. Moreover, *Erysipelatoclostridium* abundance was highly negatively correlate with colon length. In other words, BLIDF can mitigate colitis via alleviating intestinal microbiota dysbiosis.

We further investigate whether intestinal microbiota contributes to the anti-colitis effects of BLIDF in antibiotic-treated mice (Figure 5A). It can be seen in Figure 5 that BLIDF supplementation did not significantly decrease the body weight loss, DAI score, colon length shortening, and pathological damage in DSS and antibiotic-treated mice (Figure 5B–G). This result verified that the intestinal microbiota exerts a critical role in preventing colitis with BLIDF supplementation in DSS-induced colitis mice.

### 3.4. BLIDF Increased SCFAs and Secondary Bile Acids Levels in DSS-Treated Mice

SCFAs and secondary bile acids are specific classes of microbiota-derived metabolites implicated in IBD pathogenesis [30]. In agreement with decreased SCFAs levels in the IBD host [31], the present study showed a dramatic decline in the production of acetate, propionate, isobutyrate, butyrate, and valerate of cecal content in the CT+DSS group (Figure 6A). These levels were significantly restored by BLIDF supplementation. The butyrate level was re-established to that of the control group, and acetate, propionate, and valerate were elevated by 1.68-, 1.61-, and 1.51-fold in colitis mice when compared with the CT+DSS group, respectively.

Bile acid metabolism dysregulation is reported in fecal samples of IBD patients and mice [30,32]. In our work, 13 major bile acids were quantified, and the profiles of the bile acids varied significantly among the groups of mice (PERMANOVA, 999 permutations, R^2^ = 0.65996, *p* = 0.001) (Figure 6B). DSS induced a dramatic reduction in cecal contents’ primary bile acids, including tauro-α-muricholic acid (T-α-MCA), tauro-β-muricholic acid (T-β-MCA), taurocholic acid (TCA), TCDCA, α-muricholic acid (α-MCA), β-muricholic acid (β-MCA), cholic acid (CA), and chenodeoxycholic acid (CDCA), as well as the secondary bile acids hyodeoxycholic acid (HDCA), lithocholic acid (LCA), DCA, ursodeoxycholic acid (UDCA), and isolithocholic acid (isoLCA) (Figure 6C,D). This result concurs with a previous study that bile acids dramatically decreased in DSS-treated mice feces [33]. BLIDF supplementation shifted the bile acids profile in DSS-treated mice toward that of the control group (Figure 6B). It significantly decreased primary bile acids TCA and increased CDCA (Figure 6C). Notably, BLIDF supplementation altered the secondary bile acid composition. It increased the concentration of isoLCA in colitis mice to the level of the CT group and increased HDCA, LCA, and UDCA levels by 2.57-, 3.88-, and 2.04-fold, respectively, compared with the CT+DSS group (Figure 6D). These findings suggested that BLIDF supplementation can restore the DSS-induced dysregulation of microbiota-derived metabolites.

### 3.5. BLIDF Protected against DSS-Induced Intestinal Barrier Damage

The impaired gut barrier is a prominent characteristic of colitis in DSS-treated mice [34]. In our study, DSS administration induced a considerable decline in the mRNA expression of *claudin-3*, *occludin*, and *mucin2* and elevated *claudin-1* mRNA expression in colonic tissue, which were reversed by BLIDF administration (Figure 7A). Moreover, BLIDF significantly reversed the DSS-induced increase in expression of claudin-1 and the DSS-induced decrease in expression of occludin and mucin2 (Figure 7B–D). These results suggested that BLIDF supplementation likely recovered DSS-induced intestinal barrier damage by regulating the mucus and tight junction protein expressions.

### 3.6. Correlation between the Microbiota-Derived Metabolites and Intestinal Inflammatory-Related Index

Microbiota-derived metabolites, such as SCFAs and secondary bile acids were closely related to intestinal inflammation [30]. In the present work, the levels of acetate, propionate, butyrate, valerate, HDCA, and LCA exhibited significant and negative correlations with body weight loss, the pathological score, DAI score, pro-inflammatory cytokine (IL-6, TNF-α, and IL-1β) levels and claudin-1, while showing remarkable and positive correlations with colon length and mucin2 (Figure 8). And acetate, propionate, and butyrate highly positively correlated with occludin. UDCA level was highly negatively correlated with body weight loss, the pathological score, DAI score, IL-1β, and claudin-1, and highly positively associated colon length and mucin2. isoLCA exhibited significant negative correlations with body weight loss, pathological score, DAI score, pro-inflammatory cytokine (IL-6, TNF-α, and IL-1β) levels, and significant positive associated with colon length, occludin, and mucin2. These results indicate that the elevated levels of acetate, propionate, butyrate, valerate, HDCA, LCA, UDCA, and isoLCA by BLIDF supplementation in DSS-treated mice have positive effects on colitis alleviation.

## 4. Discussion

Several dietary fibers, including oligofructose-enriched inulin, resistant starch, and psyllium, can be applied as intervention therapy in IBD patients because of their role in elevating the abundance of beneficial bacteria and microbiota-derived metabolites [9,35,36,37]. However, the influence of IDF on IBD is rarely studied, likely due to its poor fermentability. Here we demonstrate that BLIDF feeding remarkably mitigated DSS-induced mice colitis via modulation of the intestinal microbiota composition. Moreover, the increased concentrations of SCFAs and secondary bile acids produced by the gut microorganisms and elevated expression levels of tight junction proteins and mucin2 probably contributed to the anti-colitis effects of BLIDF.

In the current study, dietary BLIDF dramatically reversed the DSS-induced body weight loss, increased DAI score, colon length shortening, and inflammatory lesions. Moreover, BLIDF could significantly decrease the levels of pro-inflammatory cytokines (TNF-α, IL-6, and IL-1β), and the anti-inflammatory cytokines (IL-4 and IL-10). The increased levels of IL-4 and IL-10 caused by DSS in our findings could be a triggered response to the increased expression of pro-inflammatory cytokines and severe inflammation [27]. The decreased inflammatory cytokine levels further evidenced the anti-colitis effects of BLIDF. BLIDF can also protect against DSS-induced intestinal barrier damage by increasing expression of occludin and mucin2, and decreasing expression of claudin-1. It is noteworthy that claudin-1 was positively correlated with colitis severity in IBD patients. The upregulated claudin-1 expression can inhibit the goblet cell differentiation and promote colitis-associated cancer in a Notch-dependent manner [38]. The decreased claudin-1 expression by BLIDF supplementation suggests that BLIDF can alleviate colitis severity in DSS-treated mice [39,40]. The anti-colitis effects of BLIDF were abolished when the gut microbiota was depleted by antibiotics, suggesting that the anti-colitis effects of BLIDF are highly dependent on intestinal microbiota.

The host gut harbors trillions of microbes that modulate biological processes essential for health [8]. Intestinal microbiota dysbiosis has been associated with IBD [3]. The 16S rRNA gene sequencing results demonstrated DSS-induced a significant increase in gut microbiota α-diversity, which was contrary to previous studies that α-diversity was always decreased in IBD patients [4,5]. The increased α-diversity may be due to the increased bacteria, including *Escherichia-Shigella*, *Parabacteroides*, and *Enterococcus*, which were prevalent in IBD patients [41,42]. These bacteria were not significantly decreased by BLIDF supplementation, which may have association with the unchanged gut microbiota α-diversity in DSS-treated mice. However, BLIDF significantly influenced the DSS-induced gut microbiota community change by decreasing the proportion of *Akkermansia* and increasing the proportions of *Parasutterella*, *Erysipelatoclostridium*, and *Alistipes*. *Akkermansia muciniphila* is commonly recognized as a potent anti-inflammatory intestinal bacterium [43]. However, our correlation analysis revealed that the abundance of *Akkermansia* was highly positively associated with the DAI score, the pathological score, IL-6, TNF-α, IL-1β, IL-4, IL-10, and claudin-1, and highly negatively correlated with occludin, mucin2, and colon length (Figure 4F). Accordingly, previous reports also suggested that *A. muciniphila* increased in DSS-induced colitis mice and colorectal cancer patients and was highly positively correlated with the severity of colitis [27,44]. It postulated that excessive abundance of *Akkermansia* aggravated the destruction of the intestinal barrier and exacerbated colitis in DSS-treated mice, owing to the mucin-degrading properties of *Akkermansia* [27]. Therefore, our results suggest that BLIDF feeding can significantly inhibit the DSS-induced excessive abundance of *Akkermansia* and improve the impaired intestinal barrier and gut inflammation.

Previous research indicated that the abundance of *Alistipes* decreased in IBD patients and ulcerative colitis mice [45,46] and that the addition of *Alistipes finegoldii* attenuated DSS-induced colitis [47]. In concurrence with the literature, our findings showed that *Alistipes* abundance highly negatively associated with DAI score, pathological score, and inflammatory cytokine (IL-6, TNF-α, IL-1β, and IL-10) levels, while highly positively associated with mucin2 and colon length. BLIDF supplementation significantly reversed the DSS-induced decline in the abundance of *Alistipes*.

*Erysipelatoclostridium* is a butyrate-producing bacterium [48]. It has been discovered to be enhanced in DSS-treated colitis mice [49,50]. In our study, the abundance of *Erysipelatoclostridium* was elevated in DSS-treated colitis mice, and exhibited significant positive correlations with DAI score, pathological score, TNF-α, IL-1β, IL-4, and claudin-1, and remarkable positive correlations with mucin2 and colon length. Previous studies also demonstrated that *Erysipelatoclostridium* abundance increased in DSS-treated colitis mice, while decreased in colitis alleviated mice [49,50]. However, its abundance was enhanced further by BLIDF supplementation. Hence, more investigations into the relationship between *Erysipelatoclostridium* and colitis are needed.

*Parasutterella* tends to be increased in the intestinal microbiota of IBD host [41,51], consistent with the increased abundance of *Parasutterella* induced by DSS in the present study. However, *Parasutterella* actively participates in bile acids metabolism, particularly the deconjugating process of taurine-conjugated bile acids [52]. In the present study, BLIDF feeding significantly elevated the DSS-induced increase in *Parasutterella* abundance but also decreased the DSS-induced increase in TCA (*p* < 0.05), T-α-MCA (*p* > 0.05), and T-β-MCA (*p* > 0.05). This suggested that BLIDF could alleviate colitis via modulation of the microbiota-derived metabolites. However, more evidence of the role of *Parasutterella* in bile acids regulation is needed.

The fecal metabolome can provide a functional readout of the intestinal microbial activity, which can be used as an intermediate phenotype to mediate the host-microbiome interaction [53]. Inconsistent with gut microbiota shifts, we found that BLIDF supplementation significantly restored the production of microbiota-derived metabolites, for instance, SCFAs and secondary bile acids, in colitis mice. SCFAs, particularly butyrate, promote host intestinal barrier function and alleviate inflammation via activating specific G protein-coupled receptors, inhibiting histone deacetylases [11,12,54]. Butyrate has also been shown to regulate epithelial barrier promotion and wound closure via the synaptopodin (SYNPO)-dependent pathway [55]. Our results that BLIDF feeding significantly reversed the butyrate level in colitis mice up to that of the control group was probably related to the elevated abundance of *Erysipelatoclostridium*. Moreover, our Pearson’s correlations analyses revealed significant negative correlations of acetate, butyrate, propionate, and valerate levels with body weight loss, the pathological score, DAI score, pro-inflammatory cytokine (IL-6, TNF-α, and IL-1β) levels, and claudin-1, and remarkable positive correlations of acetate, butyrate, and propionate levels with colon length, occludin, and mucin2. These trends indicated that BLIDF could alleviate DSS-induced colitis and intestinal barrier damage by enhancing butyrate, acetate, propionate, and valerate production in colitis mice.

Secondary bile acids are derived from primary bile acids (escaped from ileum) in a process dependent on biosynthetic capabilities (such as deconjugation, dihydroxylation, and desulfation) possessed by microbes [30]. In our study, the BLIDF-induced increase in deconjugated primary bile acids was probably due to the elevated *Parasutterella* abundance. The deconjugated primary bile acids are subsequently transformed into secondary bile acids by microbes such as *Bacteroides*, *Clostridium*, *Ruminococcus*, and *Lactobacillus* [30]. In the BLIDF+DSS group, we observed a dramatic enhance in HDCA, LCA, UDCA, and isoLCA levels, which were highly negatively correlated with body weight loss, pathological score, DAI score, and pro-inflammatory cytokine (IL-6, TNF-α, and/or IL-1β) levels, and highly positively associated with colon length and mucin2. Microbiota-derived bile acids can increase the number of colonic RORγ^+^ Treg cells and mitigate host susceptibility to inflammatory colitis via bile acid nuclear receptors [56]. Particularly, LCA, DCA, and UDCA have been shown to mitigate intestinal barrier damage and inflammation in animal models of colitis [23,57]. All these results above suggested that the anti-colitis effects of BLIDF were highly related to intestinal microbiota and its metabolites.

In summary, we demonstrated that BLIDF effectively mitigated DSS-induced mice acute colitis and these anti-colitis effects of BLIDF are largely depended on its modulation of the gut microbiota. Therefore, BLIDF is a promising dietary supplement targeting gut microbiota to prevent IBD prevalence. However, more research is needed to elucidate the underlying intervention mechanism linking IDF and gut microbiota modulation in the context of IBD.

## Figures and Tables

**Figure 1 nutrients-13-00846-f001:**
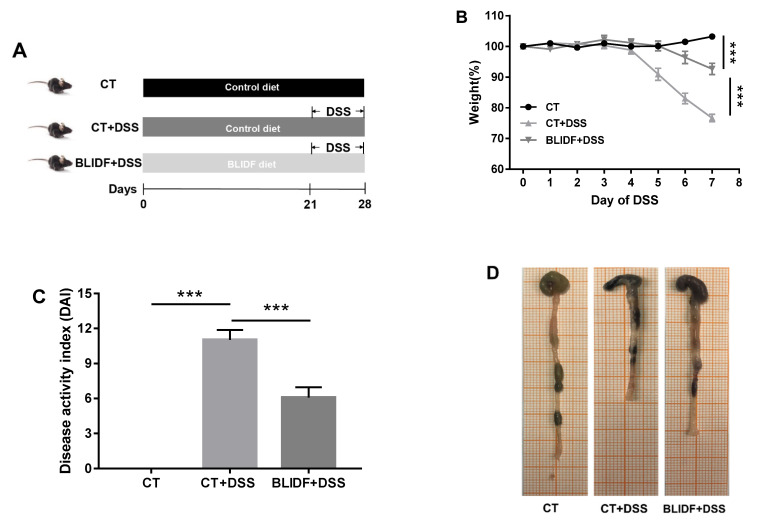
Barley leaf insoluble dietary fiber (BLIDF) mitigated dextran sulfate sodium (DSS)-induced mice colitis. (**A**) Experimental procedure of mice grouping, feeding and treatment (*n* = 8 per group); (**B**) Changes in body weight; (**C**) Disease activity index score; (**D**) Representative picture showing the overall appearance of intestinal tissue; (**E**) Colon length; (**F**) Representative H&E-stained colonic sections (10×); (**G**) Pathological scores of colonic sections. CT, control; Arrow 1 represents the crypt distortion, arrow 2 represents inflammatory cell infiltration, and arrow 3 represents goblet cell loss. ** *p* < 0.01, *** *p* < 0.001.

**Figure 2 nutrients-13-00846-f002:**
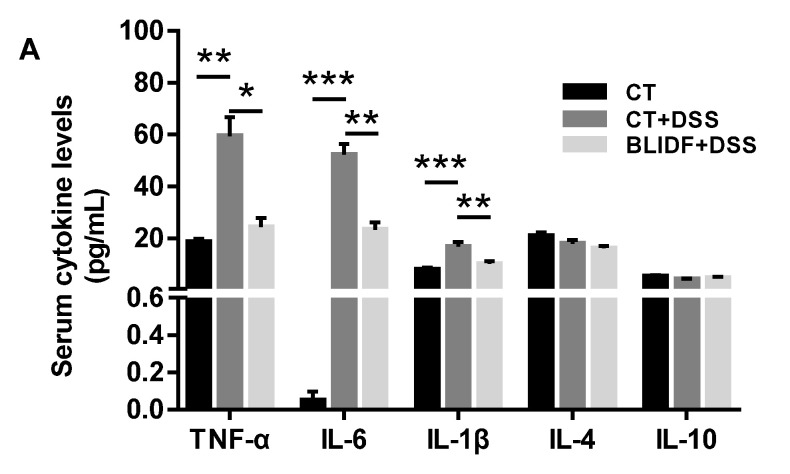
The down-regulation of inflammatory cytokine levels by BLIDF in colitis mice. The concentrations of pro-inflammatory cytokines (TNF-α, IL-6, and IL-1β) and anti-inflammatory cytokines (IL-4 and IL-10) were detected both in serum (**A**) and in colonic tissue (**B**) of mice (*n* = 8 per group). * *p* < 0.05, ** *p* < 0.01, *** *p* < 0.001.

**Figure 3 nutrients-13-00846-f003:**
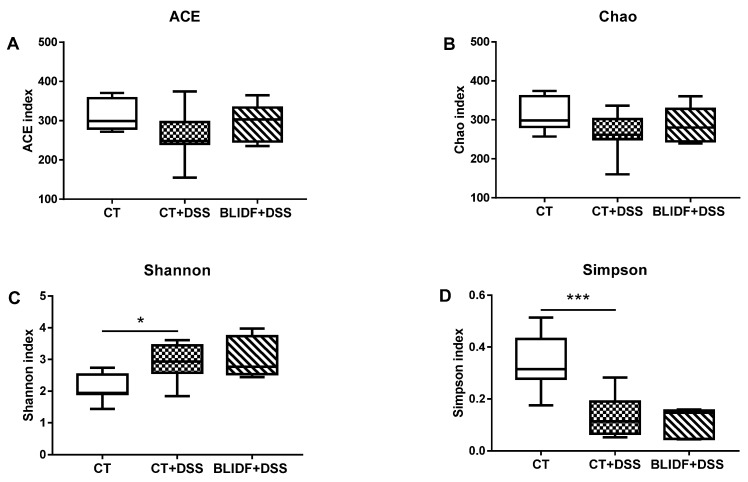
Effect of BLIDF on the bacterial α-diversity in cecal contents of colitis mice. (**A**) ACE, (**B**) Chao, (**C**) Shannon, and (**D**) Simpson index (*n* = 8 per group). * *p* < 0.05, *** *p* < 0.001.

**Figure 4 nutrients-13-00846-f004:**
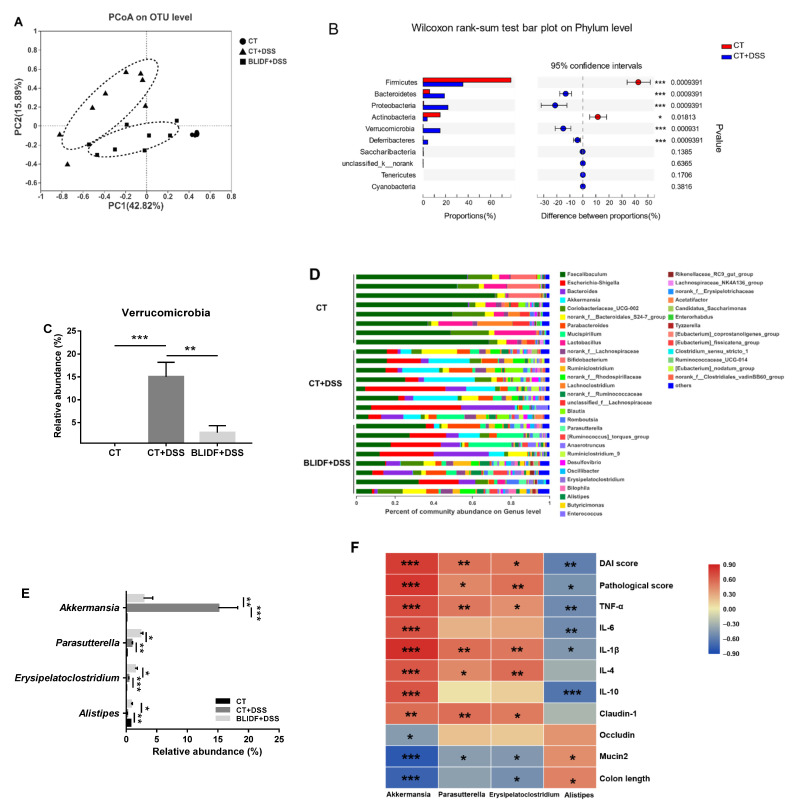
BLIDF alleviated DSS-induced gut microbiota dysbiosis. (**A**) Principal coordinate analysis (PCoA) based on Chord distance matrix at operational taxonomic units (OTU) level; (**B**) Comparison of gut bacterial composition of group CT and CT+DSS at the phylum level (determined Wilcoxon rank-sum test at FDR < 0.05); (**C**) Relative abundance of Verrucomicrobia; (**D**) Gut bacterial composition at the genus level; (**E**) Relative abundance of *Akkermansia*, *Parasutterella*, *Erysipelatoclostridium*, and *Alistipes*. (**F**) Spearman correlations analysis between the microbiota and intestinal inflammatory-related index. DAI score, Disease activity index score. * *p* < 0.05, ** *p* < 0.01, *** *p* < 0.001.

**Figure 5 nutrients-13-00846-f005:**
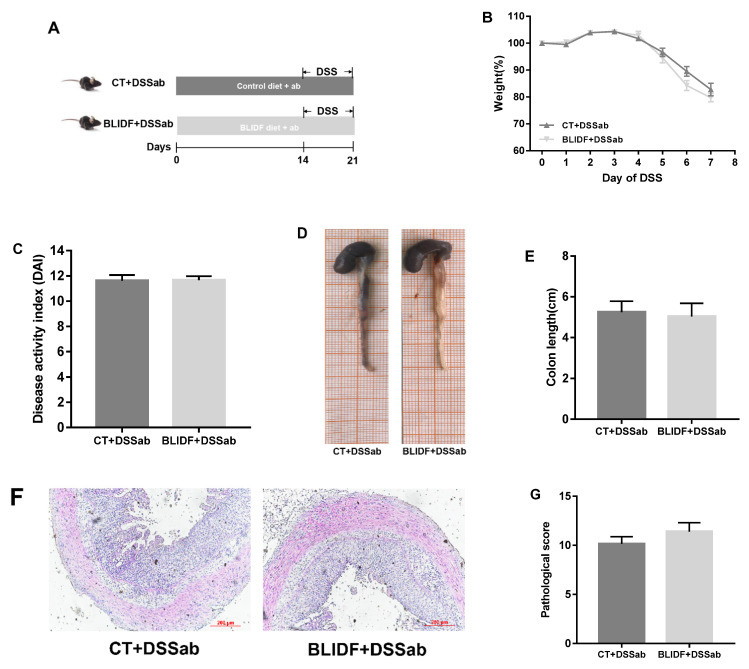
Colitis remission effects of BLIDF largely depend on gut microbiota. (**A**) Experimental procedure of mice grouping, feeding, and treatment (*n* = 8 per group); (**B**) Changes in body weight; (**C**) Disease activity index score; (**D**) Representative picture showing the overall appearance of intestinal tissue; (**E**) Colon length; (**F**) Representative H&E-stained colonic sections (10×); (**G**) Pathological scores of colonic sections. Student’s *t* test was used to test the statistical significance between the two groups.

**Figure 6 nutrients-13-00846-f006:**
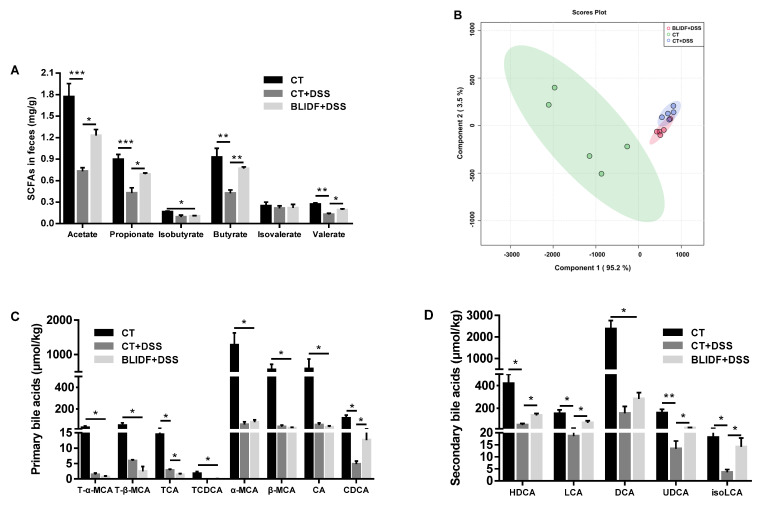
BLIDF increased SCFAs and secondary bile acid levels in DSS-treated mice. (**A**) Concentrations of SCFAs in mice cecal contents (*n* = 8 per group); (**B**) Partial Least Squares-Discriminant Analysis (PLS-DA) based on 13 bile acids detected in the mice faeces; (**C**) Concentrations of primary and (**D**) secondary bile acids. T-α-MCA, tauro-α-muricholic acid; T-β-MCA, tauro-β-muricholic acid; TCA, taurocholic acid; TCDCA, taurochenodeoxycholic acid; α-MCA, α-muricholic acid; β-MCA, β-muricholic acid; CA, cholic acid; CDCA, chenodeoxycholic acid; HDCA, hyodeoxycholic acid; LCA, lithocholic acid; DCA, deoxycholic acid; UDCA, ursodeoxycholic acid; isoLCA, isolithocholic acid. * *p* < 0.05, ** *p* < 0.01, *** *p* < 0.001.

**Figure 7 nutrients-13-00846-f007:**
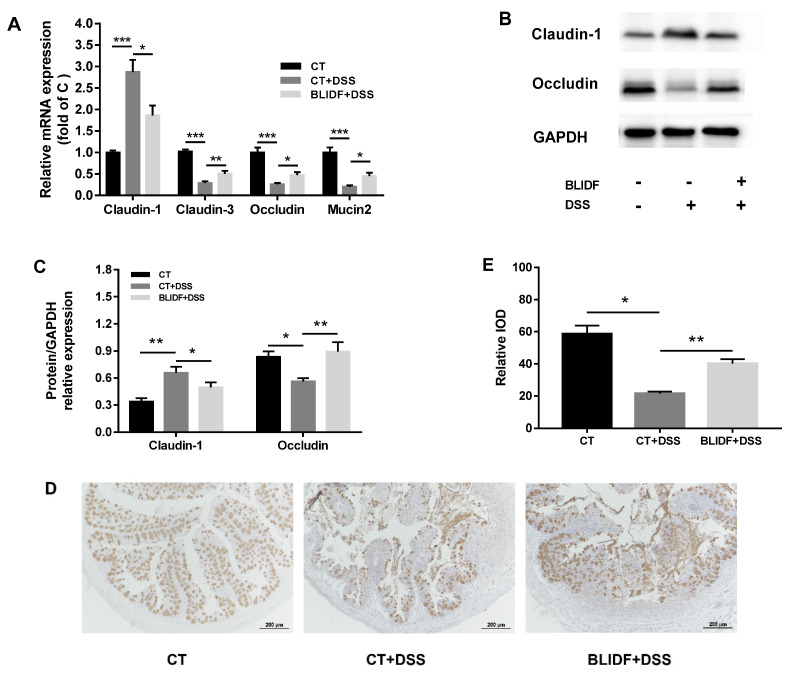
BLIDF protected against DSS-induced intestinal barrier damage. (**A**) mRNA expression of *claudin-1*, *claudin-3*, *occludin*, and *mucin2* in colonic tissue (*n* = 8 per group); (**B**) Representative western blotting of claudin-1 and occludin proteins; (**C**) Relative expression of claudin-1 and occludin proteins; (**D**) Representative immunohistochemical staining of mucin2; (**E**) Relative expression of mucin 2 protein. * *p* < 0.05, ** *p* < 0.01, *** *p* < 0.001.

**Figure 8 nutrients-13-00846-f008:**
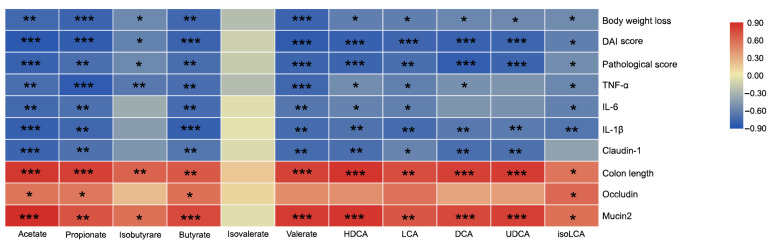
Pearson correlation analysis between the SCFAs, secondary bile acids and intestinal inflammatory-related index. DAI score, Disease activity index score. * *p* < 0.05, ** *p* < 0.01, *** *p* < 0.001.

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
