# Peer review of "Barley Leaf Insoluble Dietary Fiber Alleviated Dextran Sulfate Sodium-Induced Mice Colitis by Modulating Gut Microbiota"

_nutrients, 2021, doi:10.3390/nu13030846_

Round 1

Reviewer 1 Report

GENERAL COMMENT

In the present manuscript the Authors describe some positive effects of barley leaf insoluble dietary fiber (BLIDF), added to the diet, on body weight, cytokine levels, fecal microbiota composition, microbiota-derived metabolites, and intestinal barrier function in mice with DSS-induced colitis.

The paper presents several inaccuracies and mistakes, English language needs revision, Materials and Methods are not appropriately described, Results section contain errors, and some Results are not appropriately discussed.  Specific comments are listed below.

MAJOR REVISIONS

Abstract

-Line 15. Specify amount of IDF extract and duration of treatment.

-Line 24. Specify in which body district were these changes observed.

-Line 25. The term “elevating” is not appropriate. Please, correct.

  1. Introduction

-Line 32. Correct “demonstrates” to: “demonstrate”.

-Lines 43-55. After describing and defining the soluble fiber, make similar description and introduction for unsoluble fiber. Change the order of sentences (first description of soluble, then unsoluble). Do not describe physiological effects of IDF before its definition.

-Lines 56-57. Truncated sentence. Please, revise.

-Line 66. Change “investigate” to: “investigated”.

-Lines 71-72. Please, rephrase.

  1. Materials and Methods

In this section many important details are missing. On the basis of which rationale some methods are described here, while others are totally referred to Supplementary material? Anyway, some important informations should be present in the text-Materials and Methods section, as detailed below. From Materials and Methods the reader should have a clear idea of what has been done, which parameters analyzed, which genes, SCFAs, etc.

2.3 Animal experiments

-Line 104. Why only females? Please, justify.

-Line 107. Change “followed” to: “follows”. Was any adaptation period performed, before starting the experimental trial? Adaptation is always recommended.

-Lines 111-113. Very critical point. Which was the amount of ingested DSS? Did the Authors check the amount of drunk water? Was all the water consumed? Was water replaced during the 7 days?

-Lines 116-117. Same comment for water intake. Did the Authors check antibiotics efficacy after three days in drinking water? Which was the amount of ingested antibiotics?

-Lines 121-122. Was food intake recorded? This is very important, to correlate with weight loss, and also to know which was the effective amount of ingested BLIDF.

-Lines 124-125. Better describe all the organs/tissues collected, and how were they prepared for the analyses. What about feces for microbiota analysis? Not mentioned here. And then, I imagine that “colon segments” have been used for histological analysis, but what about tissue lysates for cytokines, etc…? These informations cannot be confined to Supplementary material.

2.5 Inflammatory cytokines analysis

-Line 137. Describe how was colon tissue prepared for cytokine analysis. There is no mention to Supplementary material here.

2.6 Quantitative real-time PCR (qRT-PCR)

-Line 142. Indicate here which “target genes” have been analyzed, and from which tissues/organs. These informations cannot be confined to Supplementary material.

2.7 16S rRNA amplicon sequencing

-Line 146. Indicate if the analysis was performed from feces or from caecal content (unclear from reading the paper). And provide here a short description of what has been analyzed (alpha diversity, beta diversity, which indices, etc.), and the software/methods used. All these informations cannot be confined to Supplementary material.

2.8 SCFAs analysis

-Line 150. Detail which SCFAs have been analyzed.

2.9 Bile acids quantification

-Lines 153-159. Why is this method well described in the text, and not demanded to Supplementary material, differently from the previous paragraphs? I think this description should be used as guide for the other paragraphs.

2.11 Statistical analysis

-Lines 167-168. Before performing the two-way and one-way ANOVA, were normal distribution and homogeneity of variance of all data checked by appropriate statistical tests? Please, specify.

  1. Results

3.1. BLIDF supplementation mitigated DSS-induced colitis in mice

-Lines 179-180. Indicate with arrows these alterations in Figure 1F.

3.3. The colitis alleviation effects with BLIDF supplementation depend on gut microbiota

-Line 215. Fecal or caecal? Please, specify here and uniform throughout the text.

-Lines 220-223. Very critical, incorrect sentence. The increased Shannon index indicates increase in alpha-diversity, not decrease! This part should be corrected, and discussed. The other indexes, shown in Figure 3, are not at all mentioned nor described.

-Line 237. Add here “as compared to control”, otherwise the sentence is unclear.

-Line 240. Correct “in DSS mice” to: “as compared to DSS mice”. In general, always indicate the terms of comparison.

-Lines 249-250. Other critical point. Did the Authors check that that animals were effectively germ free? Was antibiotics efficacy 100%? Please, describe which controls have been performed.

Discussion

Alpha-diversity data are completely missing from Discussion. The increase of alpha-diversity induced by DSS (and not decrease, as wrongly reported in the Results!) deserves a focused discussion.

Also, the data on intestinal barrier damage are not at all discussed. In particular, the Authors should provide an hypothesis for the observed increase of claudin-1 expression in DSS animals.

As well, data on pro-inflammatory cytokines should deserve more attention in Discussion.

Figures

-From the histograms it appears that only pairwise comparisons have been made (i.e. CT + DSS vs CT and BLIDF + DSS vs CT + DSS). So why in the Materials and Methods ANOVA is cited?

-Axes labelings and writings in the Figures are not readable. Please, increase font size and font type, if necessary.

Figure 1.

Panel F. Put some arrows in the photos, to indicate the alterations described in the text.

Figure 2 legend.

Reduce font size of Figure legend, similarly to the other ones.

Figure 3 legend.

Specify that is alpha diversity.

Figure 4.

Panel 4D is particularly illegible.

Panel 5D: to facilitate reading, put in the same order of treatment the histogram bars and the little legend in the Figure.

Figure 7.

C and E panels: change y axis labelling as: “Protein/GAPDH relative expression” and “Mucin2/GAPDH relative expression”.

MINOR REVISIONS

  1. Introduction

-Line 40. Change “microorganisms” to: “microbiota”.

-Line 67. Change “compositions” to: “composition” and add “intestinal” before “barrier function”.

  1. Materials and Methods

2.3 Animal experiments

-Lines 112 and 119. Change “administrated” to: “administered”.

2.10 Western blot and immunohistochemical method

Does what is indicated in brackets refer to antibodies? Please, better describe.

  1. Results

3.1. BLIDF supplementation mitigated DSS-induced colitis in mice

-Line 177. Why “apparent” body weight loss?

3.2. BLIDF decreased pro-inflammatory cytokine levels in DSS-treated mice

-Line 204. Add “findings” before “were consistent”.

3.3. The colitis alleviation effects with BLIDF supplementation depend on gut microbiota

-Line 253. Change “part” to: “role”.

Discussion

-Line 364. Change “concurrent” to: “concurrence”.

Author Response

Response to Reviewer 1 Comments

In the present manuscript the Authors describe some positive effects of barley leaf insoluble dietary fiber (BLIDF), added to the diet, on body weight, cytokine levels, fecal microbiota composition, microbiota-derived metabolites, and intestinal barrier function in mice with DSS-induced colitis.

The paper presents several inaccuracies and mistakes, English language needs revision, Materials and Methods are not appropriately described, Results section contain errors, and some Results are not appropriately discussed.  Specific comments are listed below.

Response: We sincerely appreciate the hard work and valuable suggestions of the reviewer. Below, we provided our point-by-point responses to the reviewer’s comments. We have also had our manuscript corrected by a native English speaker. The amendments of the manuscript have been marked in red.

Abstract

Point 1: -Line 15. Specify amount of IDF extract and duration of treatment.

Response 1: We sincerely appreciate the reviewer’s suggestion. The amount of IDF extract and duration of treatment was added in the manuscript and showed as follows: “In the present study, we investigated whether IDF from barley leaf (BLIDF) could inhibit gut inflammation via modulating the intestinal microbiota in DSS-induced colitis mice that fed with 1.52% BLIDF-supplemented diet for 28 days.” (Line 13-16).

Point 2: -Line 24. Specify in which body district were these changes observed.

Response 2: We sincerely appreciate the reviewer’s suggestion. Body district was specified in the manuscript and showed as follows: “Finally, BLIDF supplementation displayed an improved expression of claudin-1, occludin, and mucin2 in colons of DSS-treated mice.” (Line 24).

Point 3: -Line 25. The term “elevating” is not appropriate. Please, correct.

Response 3: We sincerely appreciate the reviewer’s suggestion and revised the term “elevating” to “increasing” (Line 25).

Introduction

Point 4: -Line 32. Correct “demonstrates” to: “demonstrate”.

Response 4: We sincerely appreciate the reviewer’s suggestion and corrected “demonstrates” to “demonstrate” (Line 32).

Point 5: -Lines 43-55. After describing and defining the soluble fiber, make similar description and introduction for unsoluble fiber. Change the order of sentences (first description of soluble, then unsoluble). Do not describe physiological effects of IDF before its definition.

Response 5: We sincerely appreciate the reviewer’s suggestion. We have changed the order of sentences and put the definition of IDF before its physiological effects as suggested by the reviewer (Line 43-57).

Point 6: -Lines 56-57. Truncated sentence. Please, revise.

Response 6: We sincerely appreciate the reviewer’s suggestion and revised the sentence as follows: “scholars recently demonstrated that cellulose supplementation could attenuate dextran sodium sulfate (DSS)-induced mice colitis, and a relationship between the cellulose-induced increase in Akkermansia muciniphila and colitis alleviation was found” (Line 57-60).

Point 7: -Line 66. Change “investigate” to: “investigated”.

Response 7: We sincerely appreciate the reviewer’s suggestion and revised “investigate” to “investigated” (Line 69).

Point 8: -Lines 71-72. Please, rephrase.

Response 8: We sincerely appreciate the reviewer’s suggestion and rephrased the sentence as follows: “This work is of great significance to evaluate the effects of IDF on intestinal microbiota composition and function in DSS-induced colitis in mice, and provide the evidence for the further application of IDF on IBD prevention.” (Line 74-76).

Materials and Methods

Point 9: In this section many important details are missing. On the basis of which rationale some methods are described here, while others are totally referred to Supplementary material? Anyway, some important informations should be present in the text-Materials and Methods section, as detailed below. From Materials and Methods the reader should have a clear idea of what has been done, which parameters analyzed, which genes, SCFAs, etc.

Response 9: We sincerely appreciate the reviewer’s suggestion. The detailed protocols have been added to the section of Materials and Methods and marked in red (Line 77-243).

2.3 Animal experiments

Point 10: -Line 104. Why only females? Please, justify.

Response 10: Thanks for the reviewer's question. In terms of the DSS-induced colitis model establish, both male and female C57BL/6J mice were widely used (1, 2). In the present study, we can evaluate the effects of BLIDF on DSS-induced colitis by comparing it to the control group. Moreover, female mice were less likely to fight (3,4), which can be easier to manage in the experiments.

References:

  1. Wyss, A.; Raselli, T.; Perkins, N.; Ruiz, F.; Schmelczer, G.; Klinke, G.; Moncsek, A.; Roth, R.; Spalinger, M.R.; Hering, L., et al. The EBI2-oxysterol axis promotes the development of intestinal lymphoid structures and colitis. Mucosal Immunol 2019, 12, 733-745, doi:10.1038/s41385-019-0140-x.
  2. Chassaing, B.; Aitken, J.D.; Malleshappa, M.; Vijay-Kumar, M. Dextran sulfate sodium (DSS)-induced colitis in mice. Curr Protoc Immunol 2014, 104, 15251-152514, doi:10.1002/0471142735.im1525s104.
  3. DeVoss, J.; Diehl, L. Murine models of inflammatory bowel disease (IBD): challenges of modeling human disease. Toxicol Pathol 2014, 42, 99-110, doi:10.1177/0192623313509729.
  4. Babickova, J.; Tothova, L.; Lengyelova, E.; Bartonova, A.; Hodosy, J.; Gardlik, R.; Celec, P. Sex Differences in Experimentally Induced Colitis in Mice: a Role for Estrogens. Inflammation 2015, 38, 1996-2006, doi:10.1007/s10753-015-0180-7.

Point 11: -Line 107. Change “followed” to: “follows”. Was any adaptation period performed, before starting the experimental trial? Adaptation is always recommended.

Response 11: We sincerely appreciate the reviewer’s suggestion and revised “followed” into “follows” (Line 112). We also performed an adaptation of 7 days before starting the experimental trials. The adaptation time was added in the manuscript and marked in red (Line 113 and Line 119).

Point 12: -Lines 111-113. Very critical point. Which was the amount of ingested DSS? Did the Authors check the amount of drunk water? Was all the water consumed? Was water replaced during the 7 days?

Response 12: Thanks for the reviewer's question. We recorded the amount of drunk water during the 7 days of DSS administration and the results were showed in the following figure (Data was shown as mean ± SED). The concentration of DSS water was 2.5% in the present study. Mouse water intake as well as the amount of ingested DSS decreased as time extended. DSS water was provided enough for mice to drunk, thus not all the water was consumed. To ensure the efficacy of DSS, DSS water was replaced every 2 days (1) (Line 118, Line 124). The figure was also added in Supplementary figure S2 (Line 127-128).

Figure 1. Mouse DSS-containing water intake. *P < 0.05, **P < 0.01, ***P < 0.001.

Reference:

  1. Chassaing, B.; Aitken, J.D.; Malleshappa, M.; Vijay-Kumar, M. Dextran sulfate sodium (DSS)-induced colitis in mice. Curr Protoc Immunol 2014, 104, 15251-152514, doi:10.1002/0471142735.im1525s104.

Point 13: -Lines 116-117. Same comment for water intake. Did the Authors check antibiotics efficacy after three days in drinking water? Which was the amount of ingested antibiotics?

Response 13: Thanks for the reviewer's question. We did not check antibiotics efficacy after three days in drinking water. Antibiotic treatment in this work referred to numerous studies (1-6). We believe this is a mature experimental method. The amount of drunk water among groups during the 7 days of DSS administration was showed in the following figure (Data was shown as mean ± SED). The water contained 2.5% DSS, 1g/L ampicillin, 1g/L metronidazole, 1g/L neomycin, and 500 mg/L vancomycin. Mouse water intake as well as the amount of ingested DSS decreased as time extended. The figure was also added in Supplementary figure S2 (Line 127-128).

Figure 2. Mouse DSS and ab-containing water intake.

Reference:

  1. Seo, S.U.; Kamada, N.; Munoz-Planillo, R.; Kim, Y.G.; Kim, D.; Koizumi, Y.; Hasegawa, M.; Himpsl, S.D.; Browne, H.P.; Lawley, T.D., et al. Distinct Commensals Induce Interleukin-1beta via NLRP3 Inflammasome in Inflammatory Monocytes to Promote Intestinal Inflammation in Response to Injury. Immunity 2015, 42, 744-755, doi:10.1016/j.immuni.2015.03.004.
  2. Garrett, W.S.; Lord, G.M.; Punit, S.; Lugo-Villarino, G.; Mazmanian, S.K.; Ito, S.; Glickman, J.N.; Glimcher, L.H. Communicable ulcerative colitis induced by T-bet deficiency in the innate immune system. Cell 2007, 131, 33-45, doi:10.1016/j.cell.2007.08.017.
  3. Kirkland, D.; Benson, A.; Mirpuri, J.; Pifer, R.; Hou, B.; DeFranco, A.L.; Yarovinsky, F. B cell-intrinsic MyD88 signaling prevents the lethal dissemination of commensal bacteria during colonic damage. Immunity 2012, 36, 228-238, doi:10.1016/j.immuni.2011.11.019.
  4. Li, Y.; Pan, H.; Liu, J.X.; Li, T.; Liu, S.; Shi, W.; Sun, C.; Fan, M.; Xue, L.; Wang, Y., et al. l-Arabinose Inhibits Colitis by Modulating Gut Microbiota in Mice. J Agric Food Chem 2019, 67, 13299-13306, doi:10.1021/acs.jafc.9b05829.
  5. Li, G.; Xie, C.; Lu, S.; Nichols, R.G.; Tian, Y.; Li, L.; Patel, D.; Ma, Y.; Brocker, C.N.; Yan, T., et al. Intermittent Fasting Promotes White Adipose Browning and Decreases Obesity by Shaping the Gut Microbiota. Cell Metab 2017, 26, 672-685, doi:10.1016/j.cmet.2017.10.007.
  6. Wang, J.; Wang, P.; Li, D.T.; Hu, X.S.; Chen, F. Beneficial effects of ginger on prevention of obesity through modulation of gut microbiota in mice. European Journal of Nutrition 2020, 59, 699-718, doi:10.1007/s00394-019-01938-1.

Point 14: -Lines 121-122. Was food intake recorded? This is very important, to correlate with weight loss, and also to know which was the effective amount of ingested BLIDF.

Response 14: We agree with the reviewer's opinion that food intake was important in weight loss and the effective amount of ingested BLIDF. We have recorded the food intake and added the data in the supplemented file. Mouse intake about 2.59 ± 0.06 control diet and 2.64 ± 0.12 g/mouse/day BLIDF diet in normal condition. Mouse intake about 1.98 mg/g body weight BLIDF per day. The amount of food intake among groups during the 7 days of DSS administration was showed in the following figures (Data was shown as mean ± SED). Mouse food intake decreased as time extended when DSS was administrated. The figure was also added in Supplementary figure S2 (Line 127-128).

Figure 3. Mouse food intake in SPF mice (C) and antibiotic-treated mice (D) under DSS administration. *P < 0.05, **P < 0.01, ***P < 0.001.

Point 15: -Lines 124-125. Better describe all the organs/tissues collected, and how were they prepared for the analyses. What about feces for microbiota analysis? Not mentioned here. And then, I imagine that “colon segments” have been used for histological analysis, but what about tissue lysates for cytokines, etc…? These information cannot be confined to Supplementary material.

Response 15: We sincerely appreciate the reviewer’s suggestion. The detailed information on tissue collection and preparation were provided and marked in red in the manuscript (Line 131-135).

2.5 Inflammatory cytokines analysis

Point 16: -Line 137. Describe how was colon tissue prepared for cytokine analysis. There is no mention to Supplementary material here.

Response 16: We sincerely appreciate the reviewer’s suggestion. The detailed preparation of colon tissue for cytokine analysis was added and marked in red in the manuscript (Line 148-153).

2.6 Quantitative real-time PCR (qRT-PCR)

Point 17: -Line 142. Indicate here which “target genes” have been analyzed, and from which tissues/organs. These informations cannot be confined to Supplementary material.

Response 17: We sincerely appreciate the reviewer’s suggestion. The detailed protocol was added in the manuscript and marked in red (Line 157-165).

2.7 16S rRNA amplicon sequencing

Point 18: -Line 146. Indicate if the analysis was performed from feces or from caecal content (unclear from reading the paper). And provide here a short description of what has been analyzed (alpha diversity, beta diversity, which indices, etc.), and the software/methods used. All these informations cannot be confined to Supplementary material.

Response 18: We sincerely appreciate the reviewer’s suggestion. The detailed protocol was added to the manuscript and marked in red (Line 167-187).

2.8 SCFAs analysis

Point 19: -Line 150. Detail which SCFAs have been analyzed.

Response 19: We sincerely appreciate the reviewer’s suggestion. The detailed protocol was added to the manuscript and marked in red (Line 189-200).

2.9 Bile acids quantification

Point 20: -Lines 153-159. Why is this method well described in the text, and not demanded to Supplementary material, differently from the previous paragraphs? I think this description should be used as guide for the other paragraphs.

Response 20: We agreed with the reviewer’s suggestion completely. We have revised the other paragraphs according to this description (Line 77-243).

2.11 Statistical analysis

Point 21: -Lines 167-168. Before performing the two-way and one-way ANOVA, were normal distribution and homogeneity of variance of all data checked by appropriate statistical tests? Please, specify.

Response 21: Thanks for the reviewer's question. Before applying ANOVA post hoc tests, the homogeneity of the variances among the groups (Levene's test) was tested. If variances are homogeneous (P ≥ 0.05), select LSD determines the significant pair(s). If variances are not homogeneous (P < 0.05), select Tamhane's T2 to determine the significant pair(s) (Line 236-240).

Results

3.1. BLIDF supplementation mitigated DSS-induced colitis in mice

Point 22: -Lines 179-180. Indicate with arrows these alterations in Figure 1F.

Response 22: We sincerely appreciate the reviewer’s suggestion. The arrows were added in Figure 1F (Line 258).

3.3. The colitis alleviation effects with BLIDF supplementation depend on gut microbiota

Point 23: -Line 215. Fecal or caecal? Please, specify here and uniform throughout the text.

Response 23: Thanks for the reviewer's question. The term here is caecal and we have uniformed the term throughout the manuscript (Line 286, Line 335, Line 365).

Point 24: -Lines 220-223. Very critical, incorrect sentence. The increased Shannon index indicates increase in alpha-diversity, not decrease! This part should be corrected, and discussed. The other indexes, shown in Figure 3, are not at all mentioned nor described.

Response 24: Thank you for pointing out the error, we sincerely apologize for our carelessly. This part was corrected and discussed in the discussion section. We also described all the indexes in Figure 3 in this part (Line 290-295).

Point 25: -Line 237. Add here “as compared to control”, otherwise the sentence is unclear.

Response 25: We sincerely appreciate the reviewer’s suggestion and the “as compared to control” was added after “Alistipes” (Line 309).

Point 26: -Line 240. Correct “in DSS mice” to: “as compared to DSS mice”. In general, always indicate the terms of comparison.

Response 26: We sincerely appreciate the reviewer’s suggestion and revised “in DSS mice” to “as compared to DSS mice” (Line 312).

Point 27: -Lines 249-250. Other critical point. Did the Authors check that that animals were effectively germ free? Was antibiotics efficacy 100%? Please, describe which controls have been performed.

Response 27: Thanks for the reviewer's question. To avoid ambiguity, we revised “germ-free-mimic mice” to “antibiotic-treated mice” (Line 328, Line 330). Our previous study demonstrated that antibiotics mixture can delete the majority of gut microbiota (1). Numerous studies also demonstrated that mice treated with a combination of vancomycin, metronidazole, neomycin, and ampicillin shown to deplete enteric microbial communities (2-4). Scholars chose antibiotic treatment to revealed the importance of gut microbiota in disease models (1, 2, 5-7). We will pay more attention to the efficacy of antibiotics in future studies, thank you very much.

Reference:

  1. Zhang, L.; Shi, M.X.; Ji, J.F.; Hu, X.S.; Chen, F. Gut microbiota determines the prevention effects of Luffa cylindrica (L.) Roem supplementation against obesity and associated metabolic disorders induced by high-fat diet. Faseb J 2019, 33, 10339-10352, doi:10.1096/fj.201900488R.
  2. Seo, S.U.; Kamada, N.; Munoz-Planillo, R.; Kim, Y.G.; Kim, D.; Koizumi, Y.; Hasegawa, M.; Himpsl, S.D.; Browne, H.P.; Lawley, T.D., et al. Distinct Commensals Induce Interleukin-1beta via NLRP3 Inflammasome in Inflammatory Monocytes to Promote Intestinal Inflammation in Response to Injury. Immunity 2015, 42, 744-755, doi:10.1016/j.immuni.2015.03.004.
  3. Garrett, W.S.; Lord, G.M.; Punit, S.; Lugo-Villarino, G.; Mazmanian, S.K.; Ito, S.; Glickman, J.N.; Glimcher, L.H. Communicable ulcerative colitis induced by T-bet deficiency in the innate immune system. Cell 2007, 131, 33-45, doi:10.1016/j.cell.2007.08.017.
  4. Kirkland, D.; Benson, A.; Mirpuri, J.; Pifer, R.; Hou, B.; DeFranco, A.L.; Yarovinsky, F. B cell-intrinsic MyD88 signaling prevents the lethal dissemination of commensal bacteria during colonic damage. Immunity 2012, 36, 228-238, doi:10.1016/j.immuni.2011.11.019.
  5. Li, Y.; Pan, H.; Liu, J.X.; Li, T.; Liu, S.; Shi, W.; Sun, C.; Fan, M.; Xue, L.; Wang, Y., et al. l-Arabinose Inhibits Colitis by Modulating Gut Microbiota in Mice. J Agric Food Chem 2019, 67, 13299-13306, doi:10.1021/acs.jafc.9b05829.
  6. Li, G.; Xie, C.; Lu, S.; Nichols, R.G.; Tian, Y.; Li, L.; Patel, D.; Ma, Y.; Brocker, C.N.; Yan, T., et al. Intermittent Fasting Promotes White Adipose Browning and Decreases Obesity by Shaping the Gut Microbiota. Cell Metab 2017, 26, 672-685, doi:10.1016/j.cmet.2017.10.007.
  7. Wang, J.; Wang, P.; Li, D.T.; Hu, X.S.; Chen, F. Beneficial effects of ginger on prevention of obesity through modulation of gut microbiota in mice. Eur J Nutr 2020, 59, 699-718, doi:10.1007/s00394-019-01938-1.

Discussion

Point 28: Alpha-diversity data are completely missing from Discussion. The increase of alpha-diversity induced by DSS (and not decrease, as wrongly reported in the Results!) deserves a focused discussion.

Also, the data on intestinal barrier damage are not at all discussed. In particular, the Authors should provide an hypothesis for the observed increase of claudin-1 expression in DSS animals.

As well, data on pro-inflammatory cytokines should deserve more attention in Discussion.

Response 28: We sincerely appreciate the reviewer’s suggestion. The increase of alpha-diversity induced by DSS, intestinal barrier damage, pro-inflammatory cytokines data, and the increased claudin-1 expression in DSS mice was discussed in this section (Line 435-447, Line 453-459).

Figures

Point 29: -From the histograms it appears that only pairwise comparisons have been made (i.e. CT + DSS vs CT and BLIDF + DSS vs CT + DSS). So why in the Materials and Methods ANOVA is cited?

Response 29: Thanks for the reviewer's question. Student's t-test, ANOVA, and ANCOVA are the statistical methods frequently used to analyze the data (1). In the present study, we referred to references (2,3) and used one-way analysis of variance (ANOVA) with least significant difference (LSD) (equal variances assumed) or Tamhane's T2 test (equal variances not assumed) to identify significant pairs.

Reference:

  1. Mishra, P.; Singh, U.; Pandey, C.M.; Mishra, P.; Pandey, G. Application of student's t-test, analysis of variance, and covariance. Ann Card Anaesth 2019, 22, 407-411, doi:10.4103/aca.ACA_94_19.
  2. Li, Q.; Zhao, H.; Zhao, M.; Zhang, Z.; Li, Y. Chronic green tea catechins administration prevents oxidative stress-related brain aging in C57BL/6J mice. Brain Res 2010, 1353, 28-35, doi:10.1016/j.brainres.2010.07.074.
  3. Yu, L.; Li, R.; Liu, W.; Zhou, Y.; Li, Y.; Qin, Y.; Chen, Y.; Xu, Y. Protective Effects of Wheat Peptides against Ethanol-Induced Gastric Mucosal Lesions in Rats: Vasodilation and Anti-Inflammation. Nutrients 2020, 12, 1-13, doi:10.3390/nu12082355.
  4. Yang, Y.; Zhang, J.; Wu, G.; Sun, J.; Wang, Y.; Guo, H.; Shi, Y.; Cheng, X.; Tang, X.; Le, G. Dietary methionine restriction regulated energy and protein homeostasis by improving thyroid function in high fat diet mice. Food Funct 2018, 9, 3718-3731, doi:10.1039/c8fo00685g.

Point 30: -Axes labelings and writings in the Figures are not readable. Please, increase font size and font type, if necessary.

Response 30: We sincerely appreciate the reviewer’s suggestion and adjusted the font size in all Figures.

Figure 1.

Point 31: Panel F. Put some arrows in the photos, to indicate the alterations described in the text.

Response 31: We sincerely appreciate the reviewer’s suggestion and the arrows have been put in Figure 1F (Line 257).

Figure 2 legend.

Point 32: Reduce font size of Figure legend, similarly to the other ones.

Response 32: We sincerely appreciate the reviewer’s suggestion and the font size of the Figure 2 legend has been reduced (Line 276).

Figure 3 legend.

Point 33: Specify that is alpha diversity.

Response 33: We sincerely appreciate the reviewer’s suggestion and revised the “bacterial diversity” to “bacterial α-diversity” (Line 335).

Figure 4.

Point 34: Panel 4D is particularly illegible.

Response 34: We sincerely appreciate the reviewer’s suggestion and have made it legible (Line 338).

Point 35: Panel 5D: to facilitate reading, put in the same order of treatment the histogram bars and the little legend in the Figure.

Response 35: Thank you for pointing out the error, we sincerely apologize for our carelessly. We have corrected the legend of panel 5D (Line 347).

Figure 7.

Point 36: C and E panels: change y axis labelling as: “Protein/GAPDH relative expression” and “Mucin2/GAPDH relative expression”.

Response 36: We sincerely appreciate the reviewer’s suggestion and revised the y-axis of Figure 7C panel as “Protein/GAPDH relative expression”. However, the levels of mucin2 expression in immunohistochemical staining were quantitated by measurement of integral optical density (IOD) (Line 400).

MINOR REVISIONS

Introduction

Point 37: -Line 40. Change “microorganisms” to: “microbiota”.

Response 37: We sincerely appreciate the reviewer’s suggestion and revised “microorganisms” to “microbiota” (Line 40).

Point 38: -Line 67. Change “compositions” to: “composition” and add “intestinal” before “barrier function”.

Response 38: We sincerely appreciate the reviewer’s suggestion. We revised “compositions” to “composition” and added “intestinal” before “barrier function” as suggested by the reviewer (Line 70).

Materials and Methods

2.3 Animal experiments

Point 39: -Lines 112 and 119. Change “administrated” to: “administered”.

Response 39: We sincerely appreciate the reviewer’s suggestion and revised “administrated” to “administered” (Line 117).

2.10 Western blot and immunohistochemical method

Point 40: Does what is indicated in brackets refer to antibodies? Please, better describe.

Response 40: Thanks for the reviewer's question. The content described in brackets refers to antibodies. We have revised and added more details to this part (Line 211-222).

Results

3.1. BLIDF supplementation mitigated DSS-induced colitis in mice

Point 41: -Line 177. Why “apparent” body weight loss?

Response 41: Thank you for pointing out the error, we sincerely apologize for our carelessly. We revised “apparent” to “significant” (Line 247).

3.2. BLIDF decreased pro-inflammatory cytokine levels in DSS-treated mice

Point 42: -Line 204. Add “findings” before “were consistent”.

Response 42: We sincerely appreciate the reviewer’s suggestion. We have revised this sentence as follows: “The trend of down-regulated pro-inflammatory and anti-inflammatory cytokine levels in colitis alleviated mice were consistent with a previous study” as suggested by another review (Line 276- Line 276).

3.3. The colitis alleviation effects with BLIDF supplementation depend on gut microbiota

Point 43: -Line 253. Change “part” to: “role”.

Response 43: We sincerely appreciate the reviewer’s suggestion and revised “part” to “role” (Line 331).

Discussion

Point 44: -Line 364. Change “concurrent” to: “concurrence”.

Response 44: We sincerely appreciate the reviewer’s suggestion and revised “concurrent” to “concurrence” (Line 475).

Reviewer 2 Report

The manuscript from Meiling Tian et. Al is exploring if barley leaf insoluble fiber can modulate microbiome composition and inflammatory response in a chemical model of colitis. The study is interesting and show promising results. However, the study suffers from the lack of negative control, regarding the effect of BLIDF or antibiotic and it is thus sometime difficult to conclude.

The method section needs more details and the discussion need to be reorganise in paragraph, rather than successive statements. Grammar can be improved.

I have the following comments:

Introduction

[30-32] IBD is not anymore considered to be an autoimmune disease and is considered as an Immune-mediated disease. Please fix it accordingly, or referred to more updated publication such as doi.org/10.1016/S0140-6736(16)31711-1 or many other, more recent work.

[32-33] Reference 2 is not describing IBD development. Please change accordingly since it is describing cross sectional case-control studies, not prospective or retrospectives cohort.

[36-38] Sentence “. Consistently, colitis developed in genetically susceptible mice with a conventional microbiota rather than under germ-free conditions”. Fix the grammar.

[41-42] Sentence “Therefore, targeting microorganisms is a novel intervention strategy to prevent IBD”, prevent IBD what? Onset? Relapse? Same issue line 44

[45-46] elevating Bifidobacterium. What does it mean ? is it abundance? Count? Activity? Please define. Same issue lines 46-48

[48-50] sentence is extremely vague, please define which SCFA you are referring to. Propionate and butyrate have different immunomodulatory properties.

In general you should stick to past tense everywhere, and not mixed present and past tense

[71-72] “The significance of this work is that it evaluates the benefits of 71 IDF on regulating intestinal microorganisms.” Function or composition both? The sentence is missing a word

Methods

Does Barley leaf powder was obtained from a single batch across all experiment? Please precise

Define AOAC, HPLC, HFK

Supplementary Table S1. Macronutrient composition of BLIDF. Add a space to the fiber type, it is misleading that the total Carbohydrate measured 88.5 are comprised of 0.3 soluble fiber + 88g of insoluble fiber. Why is there no standard deviation in this table ? Was it measured only once?

Supplementary Table S12 is the results, mean or median of BLIDF component?

For experiment I, please indicate the number of animal per cage. Please precise if multiple cage were used or not as the difference can be a cage effect and not a treatment effect.

For all experiment, is the DSS prepared daily? Is it usually an unstable component at room temperature, please add details

Mice are more nocturnal and eat more during the night. Are mice always weighted in the morning?

3000 rpm is not universal measure as it depends on the rotor. Please convert rpm to g

MAJOR issue : No CT+ab control group?

What does “ The cecal content and colon segments were collected and stored propriety for subsequent analysis” means? Appropriately?

16S rRNA gene sequencing and analysis: Is this a close referenced based OTU picking approach?

What was the sequencing depth used for rarefaction analysis?

What software was used for quality control of the 16s?

Please precise if QIIME 1 or QIIME2 was used

For the microbiome analysis, please describe what threshold is used to define significance. Were they any correction for multiple testing?

Results

Minor: Elisa [196-198] measure protein quantity, not expression

Figure 2A, why IL6 not detected in the control? If so please add in the figure legends

[204-206] comment is irrelevant to the results observed. Resveratrol is  a type of natural phenol, what is the link with BLIDF?

Does the authors measure fecal lipocalin as a direct measure of inflammation?

[224] please add the coefficient, and number of permutations used in the PERMANOVA analysis

Major:there is no CT+BLIDF control group

Figure 3B had tax that belong to Cyanobacteria. Was this not filtered out as contaminant?

Figure 3E, is this using Spearman or Pearson correlation?

Major: [250] what does “in a germ-free-mimic mice mode” mean? Is it germ-free, SPF, conventional treated with antibiotic mice? This results cannot be interpreted without a definition germ-free mimic mice model.

Again if this abx treated mice, then there is no CTAb alone tested.

Figure 5D what are you comparing ? CT+DSSab vs CT+DAAab ?? those are the same group, if this is to show that the DSSab have the same effect on all mice, move it to the supplementary material

[286] same as above add details regarding PERMANOVA

[317-319], why is the sentence relevant to this study?

Discussion

[371-373] “. This is contrary to the results that xylan butyrate ester and crude polysaccharides of Physalis pubescens L. decreased Erysipelatoclostridium in mitigating DSS-induced mice colitis” how is results with xylan comparable or relevant to BLIDF?

Supplementary Figure S3. Is this spearman correlation? This is the first-time supplementary figure 3 is describe. Move it to the results section, not in discussion

Author Response

Response to Reviewer 2 Comments

The manuscript from Meiling Tian et. Al is exploring if barley leaf insoluble fiber can modulate microbiome composition and inflammatory response in a chemical model of colitis. The study is interesting and show promising results. However, the study suffers from the lack of negative control, regarding the effect of BLIDF or antibiotic and it is thus sometime difficult to conclude.

The method section needs more details and the discussion need to be reorganise in paragraph, rather than successive statements. Grammar can be improved.

Response: We sincerely appreciate the hard work and valuable suggestions of the reviewer. Below, we provided our point-by-point responses to the reviewer’s comments. We have also had our manuscript corrected by a native English speaker. The amendments of the manuscript have been marked in red.

I have the following comments:

Point 1: [30-32] IBD is not anymore considered to be an autoimmune disease and is considered as an Immune-mediated disease. Please fix it accordingly, or referred to more updated publication such as doi.org/10.1016/S0140-6736(16)31711-1 or many other, more recent work.

Response 1: Thank you for pointing out the error, we sincerely apologize for our carelessly. We revised IBD as an immune-mediated disease and referred to the reference as suggested by the reviewer (Line 31, Line 566-569).

Point 2: [32-33] Reference 2 is not describing IBD development. Please change accordingly since it is describing cross sectional case-control studies, not prospective or retrospectives cohort.

Response 2: Thank you for pointing out the error, we sincerely apologize for our carelessly. We revised reference 2 as “Ni, J.; Wu, G.D.; Albenberg, L.; Tomov, V.T. Gut microbiota and IBD: causation or correlation? Nat Rev Gastroenterol Hepatol 2017, 14, 573-584, doi:10.1038/nrgastro.2017.88.” (Line 570-571).

Point 3: [36-38] Sentence “. Consistently, colitis developed in genetically susceptible mice with a conventional microbiota rather than under germ-free conditions”. Fix the grammar.

Response 3: We sincerely appreciate the reviewer’s suggestion. The grammar of this sentence was revised and showed as follows: “Consistently, colitis developed in genetically susceptible mice with a conventional microbiota rather than germ-free colony”. (Line 36-38).

Point 4: [41-42] Sentence “Therefore, targeting microorganisms is a novel intervention strategy to prevent IBD”, prevent IBD what? Onset? Relapse? Same issue line 44

Response 4: Thanks for the reviewer's question. we tried to express that targeting microorganism is a novel intervention strategy to prevent the pathogenesis and development of IBD. This sentence was revised and marked in red in the manuscript (Line 41-42, Line 44). 

Point 5: [45-46] elevating Bifidobacterium. What does it mean ? is it abundance? Count? Activity? Please define. Same issue lines 46-48

Response 5: Thanks for the reviewer's question. The abundance of Bifidobacterium was elevated by soluble dietary fiber. The number of Faecalibacterium prausnitzii, Ruminococcus bromii, Parabacteroides distasonis, and Clostridium leptum was elevated by Resistant starch (Line 47).

Point 6: [48-50] sentence is extremely vague, please define which SCFA you are referring to. Propionate and butyrate have different immunomodulatory properties.

Response 6: We sincerely appreciate the reviewer’s suggestion. According to the reviewer’s suggestion, the sentence was revised to “Moreover, dietary fiber altered gut microbiota and contributed to increasing the short-chain fatty acids (SCFAs) such as acetate, propionate, and butyrate [11]. Propionate and butyrate could exert anti-inflammatory effects on the intestinal epithelial cells, dendritic cells, and macrophages via inhibiting histone deacetylases activity [12]. Butyrate could also exert anti-colitis effects in mice via activating G protein-coupled receptor 43 [13]” (Line 50-55).

Point 7: In general you should stick to past tense everywhere, and not mixed present and past tense

Response 7: We sincerely appreciate the reviewer’s suggestion. We checked and revised the tense throughout the manuscript.

Point 8: [71-72] “The significance of this work is that it evaluates the benefits of 71 IDF on regulating intestinal microorganisms.” Function or composition both? The sentence is missing a word

Response 8: We sincerely appreciate the reviewer’s suggestion. According to the reviewer’s suggestion, the sentence was revised to “This work is of great significance to evaluate the effects of IDF on intestinal microbiota composition and function in DSS-induced colitis in mice, and provide the evidence for the further application of IDF on IBD prevention.” (Line 75-77).

Methods

Point 9: Does Barley leaf powder was obtained from a single batch across all experiment? Please precise

Response 9: Thanks for the reviewer's question. The barley leaf powder was obtained from a single batch across all experiments (Line 80).

Point 10:  Define AOAC, HPLC, HFK

Response 10: We sincerely appreciate the reviewer’s suggestion. the AOAC is the abbreviation of Association of Official Analytical Chemists. HPLC is the abbreviation of High-performance liquid chromatography; HFK is the abbreviation of HuaFuKang. The detailed define of AOAC, HPLC, and HFK was added in the manuscript and marked in red (Line 95-96, Line 100, Line 107).

Point 11: Supplementary Table S1. Macronutrient composition of BLIDF. Add a space to the fiber type, it is misleading that the total Carbohydrate measured 88.5 are comprised of 0.3 soluble fiber + 88g of insoluble fiber. Why is there no standard deviation in this table? Was it measured only once?

Response 11: We agreed with the reviewer’s suggestion completely, and deleted the data of total Carbohydrates. In order to make the results clearer, we combined Supplementary Table S1 and Table S2 into one table. We also measured the nutritional content of protein, fat, and moisture again, and the results were shown in Supplementary Table S1.

Point 12: Supplementary Table S12 is the results, mean or median of BLIDF component?

Response 12: Thanks for the reviewer's question. The results were the mean of the BLIDF component. We have added the footnote below the table.

Point 13: For experiment I, please indicate the number of animal per cage. Please precise if multiple cage were used or not as the difference can be a cage effect and not a treatment effect.

Response 13: We sincerely appreciate the reviewer’s suggestion. Mice were housed for 4 per cage, 2cages per group. The detailed information has been added and marked red in the manuscript (Line 115, Line 123).

Point 14: For all experiment, is the DSS prepared daily? Is it usually an unstable component at room temperature, please add details

Response 14: Thanks for the reviewer's question. we referred to the study of Chassaing, B. et al. (1) and changed DSS water every 2 days. The detailed information has been added and marked red in the manuscript (Line 119, Line 125).

Reference

  1. Chassaing, B.; Aitken, J.D.; Malleshappa, M.; Vijay-Kumar, M. Dextran sulfate sodium (DSS)-induced colitis in mice. Curr Protoc Immunol 2014, 104, 15251-152514, doi:10.1002/0471142735.im1525s104.

Point 15: Mice are more nocturnal and eat more during the night. Are mice always weighted in the morning?

Response 15: We agreed with the reviewer’s suggestion completely, and always weighted mice in the morning. Details have been added and marked red in the manuscript (Line 129).

Point 16: 3000 rpm is not universal measure as it depends on the rotor. Please convert rpm to g

Response 16: We sincerely appreciate the reviewer’s suggestion. 3000 rpm is about 1000×g in our centrifugal machine (Eppendorf, Germany) and it has been revised and marked red in the manuscript (Line 132).

Point 17: MAJOR issue: No CT+ab control group?

Response 17: Thanks for the reviewer's question. We had not done the CT+ab for 21 days. We believe that CT+DSSab and BLIDF+DSSab can support our conclusion that gut microbiota is indispensable in the anti-colitis effects of BLIDF. Moreover, numerous studies also conduct the two groups to illustrate the indispensable role of gut microbiota (1-4). For example, Zhang, L., et al. performed the groups of high fat diet+ab and high fat diet+ab+Luffa cylindrica (L.) Roem to demonstrated the indispensable role of microbiota in the anti-obesity effects of Luffa cylindrica (L.) Roem (1). Li, Y., et al. performed the groups DSS+arabinose and DSS+arabinose+ab to demonstrated the indispensable role of microbiota in the anti-colitis effects of arabinose (3). Seo, S.U. performed the groups wild mice+DSS+ab and IL1β-/- mice+DSS+ab to demonstrated that IL-1β induced by the commensal bacteria promotes colitis (3).

We conduct CT+ab for 14 days, from the figure we can observe that the overall appearance of intestinal tissue colon was the same (Figure 1A) except the enlarged cecum which is the characteristic of antibiotic-treated mice (5). We also observed that the overall appearance of colonic immunohistochemical staining of mucin2 showed no significant difference (Figure 1B), indicating that the function of goblet cell and crypt depth was not destroyed by antibiotics.

Figure 1. (A) Representative picture showing the overall appearance of intestinal tissue and representative immunohistochemical staining of mucin2 in colonic tissue.

Reference:

  1. Zhang, L.; Shi, M.X.; Ji, J.F.; Hu, X.S.; Chen, F. Gut microbiota determines the prevention effects of Luffa cylindrica (L.) Roem supplementation against obesity and associated metabolic disorders induced by high-fat diet. Faseb J 2019, 33, 10339-10352, doi:10.1096/fj.201900488R.
  2. Li, Y.; Pan, H.; Liu, J.X.; Li, T.; Liu, S.; Shi, W.; Sun, C.; Fan, M.; Xue, L.; Wang, Y., et al. l-Arabinose Inhibits Colitis by Modulating Gut Microbiota in Mice. J Agric Food Chem 2019, 67, 13299-13306, doi:10.1021/acs.jafc.9b05829.
  3. Seo, S.U.; Kamada, N.; Munoz-Planillo, R.; Kim, Y.G.; Kim, D.; Koizumi, Y.; Hasegawa, M.; Himpsl, S.D.; Browne, H.P.; Lawley, T.D., et al. Distinct Commensals Induce Interleukin-1beta via NLRP3 Inflammasome in Inflammatory Monocytes to Promote Intestinal Inflammation in Response to Injury. Immunity 2015, 42, 744-755, doi:10.1016/j.immuni.2015.03.004.
  4. Li, D.; Feng, Y.; Tian, M.; Ji, J.; Hu, X.; Chen, F. Gut microbiota-derived inosine from dietary barley leaf supplementation attenuates colitis through PPARγ signaling activation. Microbiome (accepted).
  5. Hendrickx, A.P.; Top, J.; Bayjanov, J.R.; Kemperman, H.; Rogers, M.R.; Paganelli, F.L.; Bonten, M.J.; Willems, R.J. Antibiotic-Driven Dysbiosis Mediates Intraluminal Agglutination and Alternative Segregation of Enterococcus faecium from the Intestinal Epithelium. mBio 2015, 6, e01346-01315, doi:10.1128/mBio.01346-15.

Point 18: What does “The cecal content and colon segments were collected and stored propriety for subsequent analysis” means? Appropriately?

Response 18: Thanks for the reviewer's question. The cecal contents were collected for microbiota, SCFAs, and bile acids analysis. Colon segments were divided into four parts, one part was fixed with 10% formalin and used for histological analysis. The other three parts were used for inflammatory cytokines, targeted mRNA expression, and western blot analysis, respectively. The detailed information was added and marked in red in the manuscript (Line 132-136).

Point 19: 16S rRNA gene sequencing and analysis: Is this a close referenced based OTU picking approach?

Response 19: Thanks for the reviewer's question. 16S rRNA gene sequencing and analysis is a close referenced based OTU picking approach (Line 132-136).

Point 20: What was the sequencing depth used for rarefaction analysis?

Response 20: Thanks for the reviewer's question. We obtain a total of 1,081,553 high-quality sequences and an average of 45,064.71 ± 48,46.93 sequences per sample from the 24 cecal contents samples. Rarefaction analysis indicated that adequate sequencing depth, and most bacterial diversity, were captured in the samples (Supplementary Figure S2A, B). The detail can be found in the result section (Line 176-179).

Point 21: What software was used for quality control of the 16s?

Response 21: Thanks for the reviewer's question. We using Fastp for quality filtration and merger of the 16S rRNA gene sequencing data. The detailed information was added in the part of 16S rRNA gene sequencing and analysis (Line 176-179).

Point 22: Please precise if QIIME 1 or QIIME2 was used

Response 22: We sincerely appreciate the reviewer’s suggestion. QIIME 1 was used to plot principal coordinate analysis (PCoA), and it has been revised and marked red in the manuscript (Line 176).

Point 23: For the microbiome analysis, please describe what threshold is used to define significance. Were they any correction for multiple testing?

Response 23: We sincerely appreciate the reviewer’s suggestion. For the microbiome analysis, the difference between groups was considered to be significant at p < 0.05. The principal coordinate analysis (PCoA) plot was generated in QIIME 1 using the Chord distance matrix and statistically analyzed using PERMANOVA. Differences in the relative abundance of bacteria between groups were analyzed by the Wilcoxon rank-sum test with FDR-corrected with Q < 0.05. The detailed information was added in the part of 16S rRNA gene sequencing and analysis (Line 181-185).

Results

Point 24: Minor: Elisa [196-198] measure protein, not expression

Response 24: We sincerely appreciate the reviewer’s suggestion. We have revised the “expression” to “quantity” (Line 270).

Point 25: Figure 2A, why IL6 not detected in the control? If so please add in the figure legends

Response 25: Thanks for the reviewer's question. The concentration of IL-6 in serum is low in our work. We have adjusted the figure (Line 278).

Point 26: [204-206] comment is irrelevant to the results observed. Resveratrol is a type of natural phenol, what is the link with BLIDF?

Response 26: Thanks for the reviewer's question. In this comment, we tried to express that the trend of down-regulated pro-inflammatory and anti-inflammatory cytokine levels in colitis alleviated mice were consistent with a previous study. It has been revised and marked red in the manuscript (Line 276-277).

Point 27: Does the authors measure fecal lipocalin as a direct measure of inflammation?

Response 27: Thanks for the reviewer's question. We are sorry that we did not measure fecal lipocalin. As it is during the Chinese New Year holiday, we cannot get the ELISA kit of fecal lipocalin. Moreover, inflammatory cytokines, such as IL-6, TNF-α, and IL-1β which were determined in our work was widely used to measure the inflammatory status (1-3).

  1. Li, F.; Han, Y.; Cai, X.; Gu, M.; Sun, J.; Qi, C.; Goulette, T.; Song, M.; Li, Z.; Xiao, H. Dietary resveratrol attenuated colitis and modulated gut microbiota in dextran sulfate sodium-treated mice. Food Funct 2020, 11, 1063-1073, doi:10.1039/c9fo01519a.
  2. Neurath, M.F. Cytokines in inflammatory bowel disease. Nat Rev Immunol 2014, 14, 329-342, doi:10.1038/nri3661.
  3. Koelman, L.; Pivovarova-Ramich, O.; Pfeiffer, A.F.H.; Grune, T.; Aleksandrova, K. Cytokines for evaluation of chronic inflammatory status in ageing research: reliability and phenotypic characterisation. Immun Ageing 2019, 16, 11, doi:10.1186/s12979-019-0151-1.

Point 28: [224] please add the coefficient, and number of permutations used in the PERMANOVA analysis

Response 28: We sincerely appreciate the reviewer’s suggestion. Details have been added and marked red in the manuscript (Line 297).

Point 29: Major: there is no CT+BLIDF control group.

Response 29: We sincerely appreciate the reviewer’s suggestion. We had done the CT+BLIDF control group, as can be seen in the following figure. However, in the present work, we tried to evaluate the effects of IDF on intestinal microbiota composition and function in DSS-induced colitis in mice, which can be explained using the three groups. Various studies also set similar groups to illustrate their opinion (1-3). For example, Ke, W.X. et al. set control diet group, high-fat-diet group, and high-fat-diet+ Platycodon Grandiflorus to evaluate the anti-obesity role of Platycodon Grandiflorus (1). Kim, Y. et al. set control diet+DSS, low-cellulose diet+DSS, and high- cellulose diet+DSS to evaluate the anti-colitis role of cellulose (2). Wan, P. et al. set control diet group, control diet+DSS group, and control diet+DSS+dicaffeoylquinic acids group to evaluate the anti-colitis role of dicaffeoylquinic acids.

Figure 2. BLIDF mitigated DSS-induced mice colitis. (A) experimental procedure of mice grouping, feeding and treatment; (B) Changes of body weight; (C) DAI scored in terms of body weight loss, bloody stool and colonic pathological damage; (D) Representative picture showing the overall appearance of intestinal tissue; (E) Colon length; (F) Representative H.E.-stained colon sections (10×); (G) Histopathological scores of colons sections; n = 8 per group, *P < 0.05, ** P < 0.01, *** P < 0.001.

Reference:

  1. Ke, W.X.; Bonilla-Rosso, G.; Engel, P.; Wang, P.; Chen, F.; Hu, X.S. Suppression of High-Fat Diet-Induced Obesity by Platycodon Grandiflorus in Mice Is Linked to Changes in the Gut Microbiota. J Nutr 2020, 150, 2364-2374, doi:10.1093/jn/nxaa159.
  2. Kim, Y.; Hwang, S.W.; Kim, S.; Lee, Y.S.; Kim, T.Y.; Lee, S.H.; Kim, S.J.; Yoo, H.J.; Kim, E.N.; Kweon, M.N. Dietary cellulose prevents gut inflammation by modulating lipid metabolism and gut microbiota. Gut Microbes 2020, 11, 944-961, doi:10.1080/19490976.2020.1730149.
  3. Wan, P.; Peng, Y.J.; Chen, G.J.; Xie, M.H.; Dai, Z.Q.; Huang, K.Y.; Dong, W.; Zeng, X.X.; Sun, Y. Modulation of gut microbiota by Ilex kudingcha improves dextran sulfate sodium-induced colitis. Food Res Int 2019, 126, doi:ARTN 108595

Point 30: Figure 3B had tax that belong to Cyanobacteria. Was this not filtered out as contaminant?

Response 30: Thanks for the reviewer's question. Cyanobacteria belong to prokaryotes, it will not be filtered out in 16s sequencing (Line 339).

Point 31: Figure 3E, is this using Spearman or Pearson correlation?

Response 31: Thanks for the reviewer's question. we using Pearson correlation analysis in Figure 3E, and we revised it in the legend of Figure 3 (Line 345).

Point 32: Major: [250] what does “in a germ-free-mimic mice mode” mean? Is it germ-free, SPF, conventional treated with antibiotic mice? This results cannot be interpreted without a definition germ-free mimic mice model.

Response 32: Thanks for the reviewer's question. We tried to expressed SPF mice treated with antibiotics. To avoid ambiguity, we revised “germ-free-mimic mice” to “antibiotic-treated mice” (Line 329, Line331).

Point 33: Again if this abx treated mice, then there is no CTAb alone tested.

Response 33: Thanks for the reviewer's question. We had not done the CT+ab for 21 days. We believe that CT+DSSab and BLIDF+DSSab can support our conclusion that gut microbiota is indispensable in the anti-colitis effects of BLIDF. Moreover, numerous studies also conduct the two groups to illustrate the indispensable role of gut microbiota (1-4). For example, Zhang, L., et al. performed the groups of high fat diet+ab and high fat diet+ab+Luffa cylindrica (L.) Roem to demonstrated the indispensable role of microbiota in the anti-obesity effects of Luffa cylindrica (L.) Roem (1). Li, Y., et al. performed the groups DSS+arabinose and DSS+arabinose+ab to demonstrated the indispensable role of microbiota in the anti-colitis effects of arabinose (3). Seo, S.U. performed the groups wild mice+DSS+ab and IL1β-/- mice+DSS+ab to demonstrated that IL-1β induced by the commensal bacteria promotes colitis (3).

We conduct CT+ab for 14 days, from the figure we can observe that the overall appearance of intestinal tissue colon was the same (Figure 1A) except the enlarged cecum which is the characteristic of antibiotic-treated mice (5). We also observed that the overall appearance of colonic immunohistochemical staining of mucin2 showed no significant difference (Figure 1B), indicating that the function of goblet cell and crypt depth was not destroyed by antibiotics.

Figure 1. (A) Representative picture showing the overall appearance of intestinal tissue and representative immunohistochemical staining of mucin2 in colonic tissue.

Reference:

  1. Zhang, L.; Shi, M.X.; Ji, J.F.; Hu, X.S.; Chen, F. Gut microbiota determines the prevention effects of Luffa cylindrica (L.) Roem supplementation against obesity and associated metabolic disorders induced by high-fat diet. Faseb J 2019, 33, 10339-10352, doi:10.1096/fj.201900488R.
  2. Li, Y.; Pan, H.; Liu, J.X.; Li, T.; Liu, S.; Shi, W.; Sun, C.; Fan, M.; Xue, L.; Wang, Y., et al. l-Arabinose Inhibits Colitis by Modulating Gut Microbiota in Mice. J Agric Food Chem 2019, 67, 13299-13306, doi:10.1021/acs.jafc.9b05829.
  3. Seo, S.U.; Kamada, N.; Munoz-Planillo, R.; Kim, Y.G.; Kim, D.; Koizumi, Y.; Hasegawa, M.; Himpsl, S.D.; Browne, H.P.; Lawley, T.D., et al. Distinct Commensals Induce Interleukin-1beta via NLRP3 Inflammasome in Inflammatory Monocytes to Promote Intestinal Inflammation in Response to Injury. Immunity 2015, 42, 744-755, doi:10.1016/j.immuni.2015.03.004.
  4. Li, D.; Feng, Y.; Tian, M.; Ji, J.; Hu, X.; Chen, F. Gut microbiota-derived inosine from dietary barley leaf supplementation attenuates colitis through PPARγ signaling activation. Microbiome (accepted).
  5. Hendrickx, A.P.; Top, J.; Bayjanov, J.R.; Kemperman, H.; Rogers, M.R.; Paganelli, F.L.; Bonten, M.J.; Willems, R.J. Antibiotic-Driven Dysbiosis Mediates Intraluminal Agglutination and Alternative Segregation of Enterococcus faecium from the Intestinal Epithelium. mBio 2015, 6, e01346-01315, doi:10.1128/mBio.01346-15.

Point 34: Figure 5D what are you comparing ? CT+DSSab vs CT+DAAab ?? those are the same group, if this is to show that the DSSab have the same effect on all mice, move it to the supplementary material

Response 34: Thank you for pointing out the error, we sincerely apologize for our carelessly. The right panel of Figure 5D belongs to BLIDF+DSSab. We have revised it in the figure (Line 348).

Point 35: [286] same as above add details regarding PERMANOVA

Response 35: We sincerely appreciate the reviewer’s suggestion. Details have been added and marked red in the manuscript (Line 366-367).

Point 36: [317-319], why is the sentence relevant to this study?

Response 36: We agreed with the reviewer’s suggestion completely and deleted this sentence “It is noteworthy that the increased expression level of claudin-1 may be involved in the increased risk of transformation in IBD-associated neoplasia [31].” (Line 395). Moreover, we discussed the observed increase of claudin-1 expression in DSS animals (Line 443-448).

Discussion

Point 37: [371-373] “. This is contrary to the results that xylan butyrate ester and crude polysaccharides of Physalis pubescens L. decreased Erysipelatoclostridium in mitigating DSS-induced mice colitis” how is results with xylan comparable or relevant to BLIDF?

Response 37: Thanks for the reviewer's question. We tried to express that: previously studies demonstrated that Erysipelatoclostridium abundance was increased in DSS-treated colitis mice, while decreased in colitis alleviated mice. We have revised this sentence and marked it red in the manuscript (Line 483-485).

Point 38: Supplementary Figure S3. Is this spearman correlation? This is the first-time supplementary figure 3 is describe. Move it to the results section, not in discussion

Response 38: Thanks for the reviewer's question and suggestion. This is Pearson correlation analysis in Supplementary Figure S3. We have moved it to the results section and it now showed in Figure 8 (Line 406-425).

Reviewer 3 Report

This is a very nicely written manuscript describing the effects of insoluble dietary fiber from barley leaf on the inflammatory status and gut microbiota composition of C57BL/6 mice treated with DSS to induce colitis. The authors present two very well designed mouse experiments with compelling evidence that the gut microbiota plays a role in mediating IDF effects on the prevention of colitis in mice. It was difficult to find anything to be critical of.

  1. Raw 16S rRNA sequence data should be made publicly available in a database such as the European Nucleotide Archive.
  2. Lines 12 – 14: Somewhat confusingly worded sentence
  3. Lines 38 – 42: Fecal Microbiota Transfer has been tried in human subjects with IBD with mixed results. This statement seems to ignore that body of evidence.

Author Response

Response to Reviewer 3 Comments

This is a very nicely written manuscript describing the effects of insoluble dietary fiber from barley leaf on the inflammatory status and gut microbiota composition of C57BL/6 mice treated with DSS to induce colitis. The authors present two very well designed mouse experiments with compelling evidence that the gut microbiota plays a role in mediating IDF effects on the prevention of colitis in mice. It was difficult to find anything to be critical of.

Response: We sincerely appreciate the hard work and valuable suggestions of the reviewer. Below, we provided our point-by-point responses to the reviewer’s comments. The amendments of the manuscript have been marked in red.

Point 1: Raw 16S rRNA sequence data should be made publicly available in a database such as the European Nucleotide Archive.

Response 1: We sincerely appreciate the reviewer’s suggestion. We have uploaded all raw sequence data and deposited it at the NCBI Short Read Archive database under the BioProject accession number PRJNA700444 (Line 187-188).

Point 2: Lines 12 – 14: Somewhat confusingly worded sentence

Response 2: We sincerely appreciate the reviewer’s suggestion. We revised the sentence as follows: “However, studies have paid more attention to the prevention of soluble dietary fiber against IBD via modulating gut microbiota than to the insoluble dietary fiber (IDF).” (Line 12-14).

Point 3: Lines 38 – 42: Fecal Microbiota Transfer has been tried in human subjects with IBD with mixed results. This statement seems to ignore that body of evidence.

Response 3: Thanks for the reviewer's question. Fecal microbiota transfer has mixed results in treating human IBD. However, the comment in our study is that fecal microbiota from IBD patients can promote colitis in microbiota-depleted IL-10/mice (Line 38-39).

Round 2

Reviewer 2 Report

There is still issue with grammar/English, especially in the modified sentences (text in red). Here are some examples:

"However, studies have paid more attention to the prevention of soluble dietary fiber against IBD via modulating gut microbiota than to the insoluble dietary fiber (IDF)"- “prevention of soluble dietary fiber against IBD” does not mean anything

“Barley leaf, the same batch as previously, was obtained from Hebei Biotechnology”

“The increased α-diversity may due to the increased bacteria that prevalent in IBD patients, such as Escherichia-Shigella, Parabacteroides, and Enterococcus”

Other issue:

 1) “The difference between groups was considered to be significant at p < 0.05“ But the sentence above you did correct for Q<0.5. So did consider significance at nominal p<0.05 or q<0.05 (q is a corrected p value.=)

2) Bioproject PRJNA700444 does not exist. Is it publicly available? If there is data access restriction, they should be described in the manuscript.

 3) Pearson correlations analysis between the microbiota and intestinal inflammatory-related index. If the bacteria taxa selected do not follow a normal distribution (and this usually the case), the assumption of a Pearson correlation are violated and this test cannot be used. Spearman correlation can deal with non-normal distribution. Alternatively, taxa abundance can be transform to look more normal and a Pearson correlation could be applied. The correlation results in Figure 4 are thus not reliable

Author Response

Response to Reviewer 2 Comments There is still issue with grammar/English, especially in the modified sentences (text in red). Here are some examples: Response: We sincerely appreciate the hard work and valuable suggestions of the reviewer. Below, we provided our point-by-point responses to the reviewer’s comments. We have had our manuscript corrected by a native English speaker. The amendments of the manuscript have been marked in green using the "Track Changes" function in Microsoft Word. Point 1: "However, studies have paid more attention to the prevention of soluble dietary fiber against IBD via modulating gut microbiota than to the insoluble dietary fiber (IDF)"- “prevention of soluble dietary fiber against IBD” does not mean anything Response: We agreed with the reviewer’s suggestion completely. We revised the sentence as “However, more attention has been paid to the efficacy of soluble dietary fiber than that of insoluble dietary fiber (IDF).” (Line 12-13) Point 2: “Barley leaf, the same batch as previously, was obtained from Hebei Biotechnology” Response: We sincerely appreciate the reviewer’s hard work. We revised the sentence as “Barley leaf (the same batch used in the previous study) was obtained from Hebei Biotechnology Co., Ltd. (Jiaxing, Zhejiang, China) and prepared as described previously [19].” (Line 80-81) Point 3: “The increased α-diversity may due to the increased bacteria that prevalent in IBD patients, such as Escherichia-Shigella, Parabacteroides, and Enterococcus” Response: We sincerely appreciate the reviewer’s hard work. We revised the sentence as “The increased α-diversity may be due to the increased bacteria, including Escherichia-Shigella, Parabacteroides, and Enterococcus, which were prevalent in IBD patients.” (Line 461-463) Other issue: Point 4: 1) “The difference between groups was considered to be significant at p < 0.05“ But the sentence above you did correct for Q<0.5. So did consider significance at nominal p<0.05 or q<0.05 (q is a corrected p value.=) Response: Thanks for the reviewer's question. The difference between groups was considered to be significant at q < 0.05. We sincerely apologize for our carelessness. We revised it and marked it in green in the revised manuscript (Line 185). Point 5: 2) Bioproject PRJNA700444 does not exist. Is it publicly available? If there is data access restriction, they should be described in the manuscript. Response: Thanks for the reviewer's question. We have uploaded all raw sequencing data and deposited it at the NCBI Short Read Archive database under the BioProject accession number PRJNA700444. Details can be found in the following email content. The release date is 2022-03-08. We have added the release date in the manuscript (Line 188). Point 6: 3) Pearson correlations analysis between the microbiota and intestinal inflammatory-related index. If the bacteria taxa selected do not follow a normal distribution (and this usually the case), the assumption of a Pearson correlation are violated and this test cannot be used. Spearman correlation can deal with non-normal distribution. Alternatively, taxa abundance can be transform to look more normal and a Pearson correlation could be applied. The correlation results in Figure 4 are thus not reliable Response: Thank you for pointing out the error, we sincerely apologize for our carelessness. We use the SPSS and found that not all bacteria taxa selected follow a normal distribution. We use spearman correlation analysis to show the relationship between the microbiota and intestinal inflammatory-related index. We have revised the related text content throughout the manuscript and marked in green. (Line 323-331, Line 471, Line 482-485, and Line 489-491)
